# Shark mandible evolution reveals patterns of trophic and habitat-mediated diversification

Faviel A. López-Romero [1,2✉], Sebastian Stumpf [1], Pepijn Kamminga[3], Christine Böhmer [4,5,6], Alan Pradel[7], Martin D. Brazeau [8,9] & Jürgen Kriwet [1,2]

Environmental controls of species diversity represent a central research focus in evolutionary biology. In the marine realm, sharks are widely distributed, occupying mainly higher trophic levels and varied dietary preferences, mirrored by several morphological traits and behaviours. Recent comparative phylogenetic studies revealed that sharks present a fairly uneven diversification across habitats, from reefs to deep-water. We show preliminary evidence that morphological diversification (disparity) in the feeding system (mandibles) follows these patterns, and we tested hypotheses linking these patterns to morphological specialisation. We conducted a 3D geometric morphometric analysis and phylogenetic comparative methods on 145 specimens representing 90 extant shark species using computed tomography models. We explored how rates of morphological evolution in the jaw correlate with habitat, size, diet, trophic level, and taxonomic order. Our findings show a relationship between disparity and environment, with higher rates of morphological evolution in reef and deep-water habitats. Deep-water species display highly divergent morphologies compared to other sharks. Strikingly, evolutionary rates of jaw disparity are associated with diversification in deep water, but not in reefs. The environmental heterogeneity of the offshore water column exposes the importance of this parameter as a driver of diversification at least in the early part of clade history.

[1] University of Vienna, Faculty of Earth Sciences, Geography and Astronomy, Department of Palaeontology, Evolutionary Morphology Research Group, Josef-Holaubek-Platz 2, 1190 Vienna, Austria. [2] University of Vienna, Vienna Doctoral School of Ecology and Evolution (VDSEE), Djerassiplatz 1, 1030 Vienna, Austria. [3] Naturalis Biodiversity Center, Darwinweg 2, 2333 CR Leiden, The Netherlands. [4] MECADEV UMR 7179 CNRS/MNHN, Département Adaptations du Vivant, Muséum National d'Histoire Naturelle, CP 55, 57 rue Cuvier, 75231 Paris, France. [5] Department für Geo- und Umweltwissenschaften und GeoBio-Center, Ludwig-Maximilians-Universität München, Richard-Wagner-Straße 10, 80333 München, Germany. [6] Zoologisches Institut, Christian-Albrechts-Universität zu Kiel, Am Botanischen Garten 1-9, 24118 Kiel, Germany. [7] CR2P, Centre de Recherche en Paléontologie – Paris, Muséum National d'Histoire Naturelle—Sorbonne Université—CNRS, CP 38, 57 rue Cuvier, F75231 Paris, Cedex 05, France. [8] Department of Life Sciences, Imperial College London, Silwood Park Campus, Ascot, SL5 7PY London, UK. [9] The Natural History Museum, Cromwell Road, London SW7 5BD, UK. ✉email: faviel.l.r@gmail.com

Sharks (selachimorph elasmobranchs) are a globally distributed group of predatory fishes in marine and rarely freshwater environments. They comprise more than 500 living species occupying mainly higher trophic levels[1–4]. Falling into two major groups, the Galeomorphii (e.g., mackerel, ground and carpet sharks) and the lesser-known Squalomorphii (i.e., cow, frilled, dogfish, saw and angel sharks), modern sharks have developed a wide range of lifestyles and expanded into various ecological niches since their first appearance in the fossil record during the Early Jurassic[2,5,6].

Sharks are an attractive target for macroevolutionary studies of marine vertebrates. A rich body of literature focuses on teleosts in proximity to reefs[7–11]. Often these studies seek to reveal how environments and innovations interact to drive diversification in specific clades. However, teleost ecological and evolutionary dominance has frequently been attributed to several key innovations and probably also events such as genome duplication[12–14]. Furthermore, with nearly 35,000 living species[15], teleosts represent a vast taxonomic inventory that is difficult to comprehensively sample, compounded with deep uncertainty about the clade's deepest phylogenetic branches;[16–19] but see[20]. Sharks, by contrast, comprise approximately 500 species—a much more tractable group for high-density sampling. Sharks also lack some of the specific novelties of the feeding system and genome that have been attributed to the species richness of teleosts[15,21,22]. Thus, macroevolutionary studies of sharks can provide a useful contrast to teleost-based studies, allowing us to identify common factors promoting differential diversification rates over time.

Sharks and fishes generally are important targets for deep-time macroecological and evolutionary study. Recent studies of fish diversification in both the neontological and palaeontological records have revealed striking exceptions to well-established biodiversity gradients (such as onshore-offshore diversification, the origination of higher taxa preferably on nearshore environments, later expanding offshore in their evolutionary history[23] and latitudinal biodiversity gradients) in various groups of fishes and fish-like early vertebrates[24–27]. Sorenson et al.[24], for instance, found equivocal results for an onshore-offshore diversification gradient in sharks. More pointedly, Martinez et al.[28] identified the deep-water realm as a crucial hotspot for morphological diversity of bony fishes. These studies show that sharks are potentially an exception to the classic onshore-offshore diversification gradient and, that morphological parameters relating to lifestyle and feeding provide a more nuanced view of diversification along environmental gradients than pure speciation rate or species richness.

The purpose of this paper is two-fold: assess the assumption of ecological signal in mandible shape and use these results to explore the evolutionary dynamics of trophic evolution. Lower jaw morphology is frequently used as a proxy for functional and ecological space occupation in the fossil[29–31] and neontological record of fishes[9,10,32–34]. The jaw apparatus of sharks display several important morphofunctional modifications[21,35–39] that enable specialised feeding behaviours, such as filter feeding, durophagy, and piscivory[40–44]. Prey capture and processing in sharks is reflected in their jaw suspension[36,37,45,46] and biting mechanisms[47–51]. However, palaeobiological studies rarely possess direct data on trophic ecology that can validate underlying assumptions[52]. The trophic ecological variations among extant sharks have been thoroughly studied, with several surveys examining their stomach contents (e.g.[1,53]), but also via stable isotopes analyses (e.g.[52,54,55]) leading to the characterisation of trophic levels they occupy, making it possible to validate the assumptions of previous works. We use comparative phylogenetic techniques to explore how morphological disparity in the feeding system evolved in relation to habitats and provide novel information about potential biotic and abiotic drivers of biodiversity 'hotspots'.

## Results

**Jaw shape variation.** The principal components analysis (PCA) of all specimens (Fig. 1) allows an exploration of major shape variations in shark mandibles. Examining the relative warps plots (Fig. 1, Supplementary Fig. S7) reveals the major landmark variation on Procrustes that characterise extreme members of each axis. The negative extreme for PC1 (40.9% of the total variation) is characterised by slender, anteriorly tapering jaws; the dental groove extends most of the length of the jaw; the articular condyle is relatively narrow and gracile. The positive extremes of PC1 are represented by specimens with dental grooves roughly co-equal to jaw length or shorter than the articular/adductor insertion region (Supplementary Fig. S1 and S6 for species labels). The articular condyle in these representatives is wide relative to jaw length and extremely robust. In PC2 (15.75% of the variation), positive values correspond to a dorsoventrally deep mandibular symphysis and a posteriorly deep dental groove. In PC2, negative scores correspond to a broader adductor insertion region in lateral view and low dental groove. In the positive scores, the dental groove shows a prominent widening, which extends posteriorly, leading to a short posterior profile in the lateral view. Relative warps show that variation in PC3 (12.09% of the variation) is related to an expanded adductor insertion area in lateral view that tapers anteriorly towards the symphysis; these jaws are overall more 'J-shaped', with the articular sitting approximately level with the front of the tooth row. In the negative scores of PC3, the jaws are elongated, with a low profile in the posterior region, and the dental groove is straighter in the dorsal view as opposed to positive PC3 scores (Supplementary Fig. S7).

Most carcharhiniforms and lamniforms are clustered within a narrow area in the morphospace compared to squaliforms, which occupy an extended area of the morphospace. Meanwhile, the orectolobiforms diverge from the main galeomorph aggregation. The other analysed shark orders (Echinorhiniformes, Pristiophoriformes, Squatiniformes, Hexanchiformes, and Heterodontiformes), which are represented by fewer species, are mostly located within the main cluster along with lamniforms and carcharhiniforms. The remaining PCs represent less than 10% of the variation and display more redundancy (Supplementary Fig. S8). Overall, species of the same order cluster together in jaw shape space, which also is reflected in the phylogenetic signal ($K_{mult} = 0.5318$, $p = 0.001$) and in the phylomorphospace (Fig. 2a).

When considering habitat, the jaw phylomorphospace displays a consistent separation between groups (Fig. 2b). Accordingly, most of the reef-associated species are located around positive scores of the phylogenetically aligned principal component (PaPC) PaPC1 (70.88%), while pelagic species are restricted to negative scores of PaPC1, which is also shared by shelf-living species. Interestingly, deep-sea species are spread across the extremes of PaPC1 and mainly positive PaPC2 scores in the phylomorphospace. Many species included in the deep-sea group are mainly squaliforms, hexanchiforms, some carcharhiniforms (catsharks) and one lamniform, *Mitsukurina owstoni* (Goblin shark).

The trophic level (TR) of each species analysed ranges from TR = 3.2 for the filter feeder *Cetorhinus maximus* (Basking shark) to TR = 4.6 for many top predatory species. We divided the values of trophic levels into three categories (see Material and methods). Accordingly, the phylomorphospace shows a considerable overlap between low-level (LP, TR = 3.2–3.8) and meso-level

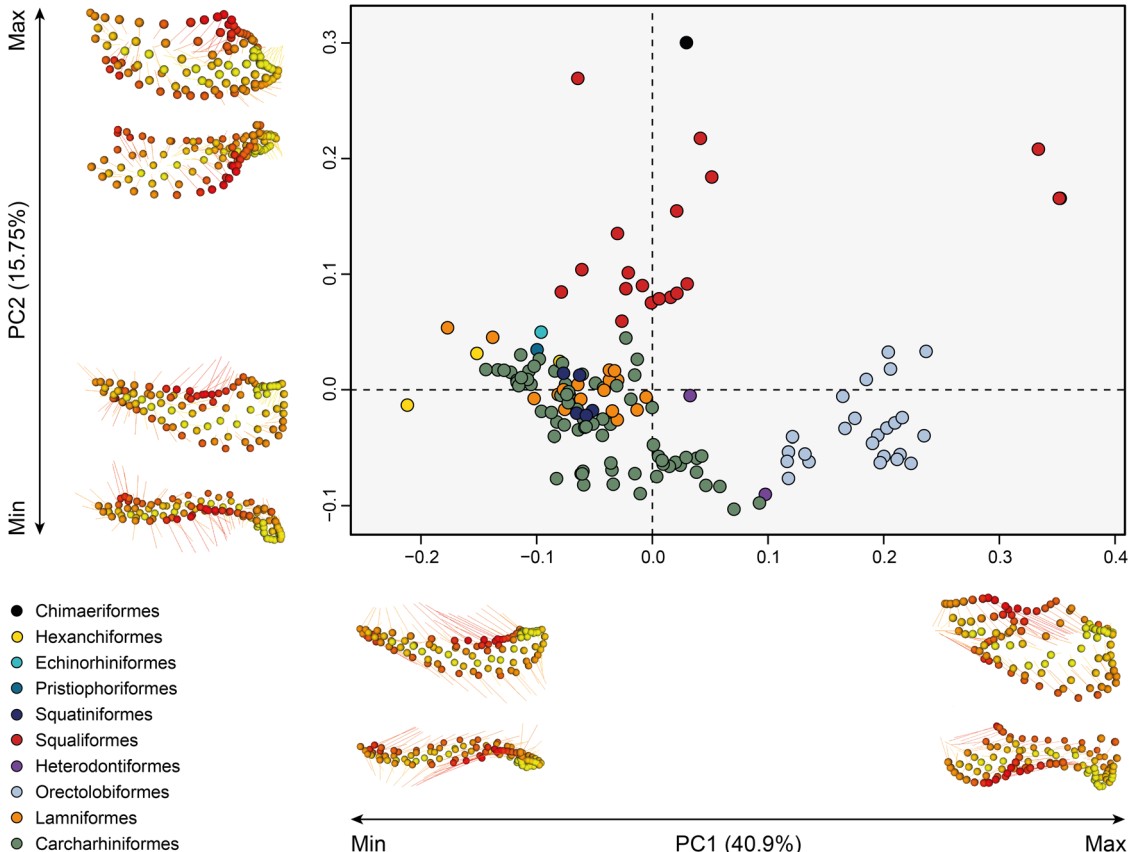

**Fig. 1 Shape variation of the lower jaw among 145 specimens (90 species) of chondrichthyans.** The first two PCs are shown with their explained proportion of variance. The groups are displayed by coloured points corresponding to their respective order, as indicated by the colour code. Chimaeriformes ($n = 1$); Hexanchiformes ($n = 3$); Echinorhiniformes ($n = 1$); Pristiophoriformes ($n = 2$); Squatiniformes ($n = 5$); Squaliformes ($n = 19$); Heterodontiformes ($n = 2$); Orectolobiformes ($n = 24$); Lamniformes ($n = 17$); Carcharhiniformes ($n = 71$), number of specimens between brackets. Landmark variations between the maximum and minimum for each PC are displayed along the axes.

predators (MP; TR = 3.81–4.2) (Fig. 2c). The top predators are in the positive PaPC2 scores, LPs and MPs widely overlap along PaPC1 and PaPC2. Interestingly, some extreme shapes are included in the category of top predators, like the squaliforms *Isistius brasiliensis* (Cookiecutter shark) and *Centroscymnus coelolepis* (Portuguese dogfish), because of their higher trophic level index of TR = 4.2.

The clustering analysis of the stomach content categories suggests eight major feeding guilds (Supplementary Figs. S3 and S4). The prey content groups display a considerable overlap in the phylomorphospace. The FISH (fish consumers) group is particularly widespread in the phylomorphospace (Fig. 2d), but more species are within negative PaPC1 and positive PaPC2 scores encompassing mostly carcharhiniforms and lamniforms. The CEPH (cephalopod consumers) and CR (crustacean consumers) groups display a more expanded phylomorphospace occupation, similar to the pattern seen in generalists. The BP (big predators) group is clustered in a narrow portion of the phylomorphospace and is composed of only carcharhiniforms and lamniforms. Notably, the groups INV (invertebrate consumers), MOLL (hard-shelled molluscs consumers), and ZOO (zooplankton consumers) occupy extreme values of positive PaPC1 and negative PaPC2, which are represented by few species, like the *Oxynotus centrina* (Angular roughshark) and *Hemiscyllium trispeculare* (Speckled carpetshark) in the INV group, and the MOLL group consisting of *Stegostoma fasciatum* (Zebra shark) and *Heterodontus francisci* (Horn shark).

**Jaw shape differences depend on order, diet, trophic level and habitat.** The phylogenetic MANOVA reveals only habitat as a significant predictor of shape (Table 1). On the other hand, when size is considered using a phylogenetic MANCOVA of shape on log(centroid size) (as an index for body size) and each of the categories, we found significant differences among orders, trophic levels, diet composition, and habitat occupation (Table 1). The regression of shape on log(centroid size) shows the lamniforms and carcharhiniforms are among the largest species, while squaliforms and orectolobiforms are the smallest (Supplementary Fig. S9a; Supplementary Table S10). A similar pattern is observed when the shape is regressed on log(centroid size) and trophic level, with most of the low-level predators and mesopredators displaying small size and top predators larger size overall (Supplementary Fig. S9b). Likewise, among the diet content groups, a regression of shape on log(centroid size) shows the largest species within BP and FISH groups, while specimens included in the MOLL, INV and CR groups are among the smallest (Supplementary Fig. S9c). Finally, the regression for the habitat groups shows most of the larger species are within the pelagic group, with a few deep-sea species (*Chlamydoselachus anguineus*, *Hexachus griseus*, *Heptrachias perlo*, *Echinorhinus brucus*), while most of the smaller species are found within the deep sea and reef-associated assemblages (Supplementary Fig. S9d).

**Mean Evolutionary Rates and Disparity.** Comparison of the mean evolutionary rate of the lower jaw between orders shows

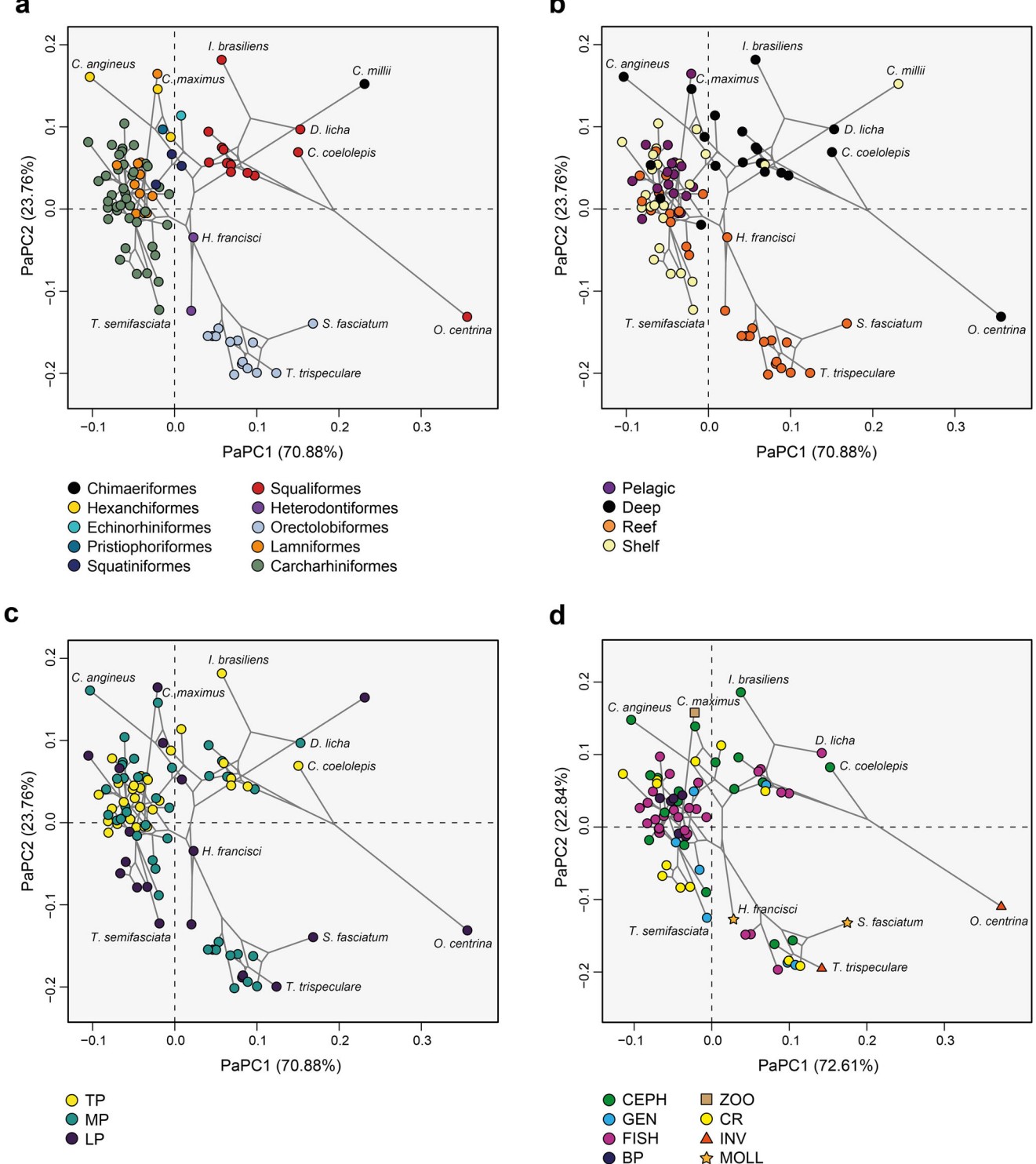

**Fig. 2 Phylomorphospaces of the lower jaw shape with averaged values for the species for the first two PaPCs. a** Orders, **b** habitat, **c** trophic level and **d** diet content. The colour code indicates the corresponding group, abbreviations as follows: BP big predator, CEPH cephalopod consumer, CR crustacean consumer, FISH fish consumer, GEN generalist, INV invertebrates consumer, MOLL molluscs consumer (hard-shelled), ZOO zooplankton consumer, LP low-level predator, MP mesopredator, TP top predator.

that orectolobiforms have a higher rate than the remaining orders ($\sigma = 1.439775e^{-06}$), followed by squaliforms and hexanchiforms ($\sigma = 1.408004e^{-06}$, 1.022 fold change; and $\sigma = 1.136257e^{-06}$, 1.267 fold change respectively), but compared to Lamniformes and Carcharhiniformes, the difference is more evident (7.287 and 12.989 fold change respectively) (Fig. 3a) (Supplementary

Table S5). When analysing the taxa divided into specific prey categories, we observe that members of the CEPH group evolved faster than all other groups ($\sigma = 3.617352e^{-07}$), while the GEN group has the lowest rate ($\sigma = 1.366694e^{-07}$, 2.646 fold change). The groups subdivided by trophic levels show that low-level predators evolved faster ($\sigma LP = 3.436747e^{-07}$), followed by top

**Table 1 Results of the pMANOVA for the factors interacting with centroid size.**

|  |  | Wilks' Λ | Pagel's λ |
|---|---|---|---|
| Orders | CS | 0.8729 *** |  |
|  | Ord | 4.4554 *** |  |
|  | CS:Ord | 4.2802 *** |  |
| Habitat | CS | 0.8624 *** |  |
|  | Hab | 2.6393 *** |  |
|  | CS:Hab | 2.6358 *** | 0.9935 |
| Trophic Level | CS | 0.8432 *** |  |
|  | TrL | 1.6709 *** |  |
|  | CS:TrL | 1.6803 * | 0.9977 |
| Diet | CS | 0.9404 *** |  |
|  | Diet | 3.6452 *** |  |
|  | CS:Diet | 3.4959 ** | 0.8578 |
|  | Hab | 2.358 ** | 0.7141 |
|  | Ord | 3.749 | 0.1987 |
|  | TrL | 1.528 | 0.7021 |
|  | Diet | 3.173 | 0 |

Significance based on permutations ($n = 999$). Wilkis' Λ value indicated $P$ values significant at alpha levels: *≤0.05, **≤0.01, ***≤0.001, and Pagel's λ

($\sigma TP = 2.254861e^{-07}$, 1.524 fold change) and meso-predators ($\sigma MP = 1.80186e^{-07}$, 1.907 fold change). Finally, the comparison of evolutionary rates by habitat shows that deep-sea species evolved faster than the rest ($\sigma = 9.40033e^{-07}$, 1.419 fold change in relation to the reef; 5.251 fold change in relation to pelagic, and 7.692 fold change in relation to shelf). But reef-associated species also display high evolutionary rates ($\sigma = 6.624241e^{-07}$) when compared to shelf and pelagic species (Fig. 3a). The estimation of evolutionary rates at single landmarks, among the four main orders (Squaliformes, Orectolobiformes, Carcharhiniformes, and Lamniformes) shows that the highest rates are localised in the dental groove for squaliforms, while orectolobiforms have higher rates at the mandibular knob-palotoquadrate articulation and sustentaculum (Supplementary Fig. S10). Finally, both carcharhiniforms and lamniforms display similar rates, especially at the symphysis, posterior edge of the dental groove, and the lower margin curve (Supplementary Fig. S10).

The results of the morphological disparity, as Procrustes variance (PV), indicate significant differences in the jaw shape diversity between orders (Fig. 3b). Squaliform sharks display the highest disparity compared to the remaining orders (PV = 0.036, p < 0.001), but orectolobiforms also show differences to the other orders (PV = 0.016, $p$ < 0.001). Only carcharhiniforms and lamniforms show no difference from each other (PV = 0.011; PV = 0.01, $p$ = 0.6217) (Supplementary Table S6). Between the dietary groups, CEPH displays the highest disparity (PV = 0.04), while BP has the lowest value (PV = 0.008); all the pairwise comparisons between groups are significant ($p$ < 0.001) (Supplementary Table S7). When the species were compared as representing trophic level groups, the low-level predators had the highest disparity, followed by meso- and top predators (PV = 0.044; 0.03, 0.019, respectively) ($p$ < 0.001) (Supplementary Table S8). Finally, the disparity between habitats shows that deep-sea species have the highest disparity (PV = 0.042), followed by reef (PV = 0.026), shelf (PV = 0.015) and pelagic species (PV = 0.01), with significant differences between all groups ($p$ < 0.001) (Supplementary Table S9).

**Ancestral state reconstruction of trophic ecology.** The prey content ancestral state reconstruction indicates that different strategies have evolved independently multiple times within the orders (Fig. 4a). The majority of species have a piscivorous diet,

most likely representing the ancestral state for the big predatory species. Also, the evolution towards large predators occurred independently several times in both carcharhiniforms and lamniforms. Other specialisations, like feeding on hard-shelled prey (either on crustaceans or molluscs), represent independent events, such as among hammerhead sharks and particularly among species of the genus *Mustelus*. Regarding the habitat, there is a consistent pattern for the Squaliformes in deep-sea habitats, with an inferred ancestral state for that trait (Fig. 4b). Similarly, the Orectolobiformes display a consistent pattern for reefs and its ancestral state as well. Only the Carcharhiniformes display a wider diversity of habitats, which accordingly might have evolved from a shelf distribution, and from this inferred shelf habitat, the Lamniformes later transitioned to a pelagic habitat (Fig. 4b).

The evolutionary rates reconstructed in Bayes Traits indicate several branch-specific higher rates across the phylogeny (Fig. 5a). Overall, in galeomorphs rates are lower than in squaliomorphs. Orectolobiforms, however, display a notably increased rate at their base, which is consistent with the mean evolutionary rate by groups (Fig. 3a). Other cases within galeomorphs, like *Cetorhinus maximus* (Basking shark), display a higher evolutionary rate than the other Lamniformes. Within squaliomorphs several more instances of elevated evolutionary rates are found, especially species like *Oxynotus centrina*, *Dalatias licha*, *Isistius brasiliensis* and *Centroscymnus coelolepis* display extreme shapes in the previous analyses (Figs. 1 and 2a). All these species are characterised by a wide spectrum of prey consumption, ranging from invertebrates to cephalopods and even large mammals, and occur in the deep sea. The disparity through time analysis indicates that the morphological disparity follows an early shape high disparity, which declines through time (Fig. 5c), and this pattern is also seen from the 100 randomly sampled trees. This also is supported by estimating the model of trait evolution for the whole set of landmarks, where Early Burst is the best-supported model (GIC = −770590.6, Supplementary Table S11). We further explored shifts in evolutionary rates depending on distinct regimes. The results indicate differences in the posterior distribution of the parameters only for the habitat comparisons (Supplementary Tables S12–S14; Supplementary Fig. S11). The pairwise comparison in the rates shifts reveals that only deep-sea species display differences compared to pelagic, shelf and reef species. A possible shift in rate was observed in trophic levels, in particular for low-level predators. However, this shift was no longer supported when the analysis was run on the 100 randomly chosen trees (Supplementary Table S12).

Finally, we estimated the ancestral lower jaw shape for selachimorph elasmobranchs based on all landmarks and including all species. The chimaeriform *Callorhinchus milii* was included as an outgroup to examine shape departure from selachimorphs (Supplementary Fig. S12). The resulting estimation suggests a jaw with a relatively high anterior symphysis and deeper dental groove, which nevertheless is shorter than the extreme shapes found in squaliforms. A strong posterior curvature on the attachment area for the adductor mandibulae also is present. The quadrato-medial and quadrato-lateral joints and mandibular knob are well developed, although it is not as hypertrophied as in orectolobiforms, or reduced as in hexanchiforms.

## Discussion

Mandible shape variation in sharks can be explained in terms of functional interpretations of the major shape differences. This helps validate assumptions that studies of shape variation in jaw structure can provide reasonable proxies for primary feeding ecology, as is commonly done in palaeobiological and

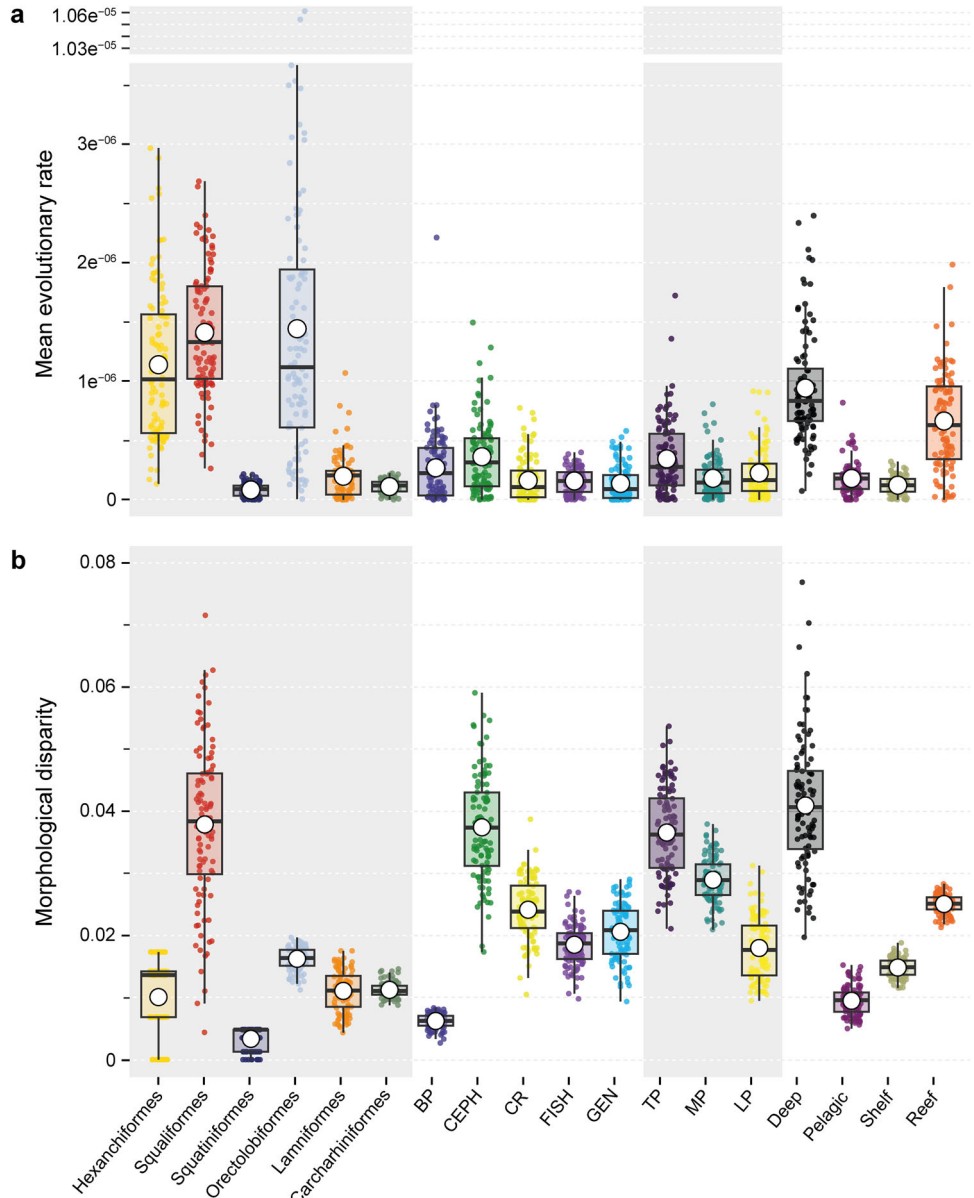

**Fig. 3 Mean evolutionary rates and the morphological disparity between orders, diet guilds, trophic level, and habitat. a** Evolutionary rate for each group ($\sigma$) estimated from the stochastic mapping of each categorical variable over 100 simulations. **b** Morphological disparity as Procrustes variance, comparison between the groups. A large white dot in the middle of the boxes shows the mean, while the lines inside the box show the median. Abbreviations as in Fig. 2.

macroecological studies[30,56–59]. The jaw-closing mechanical advantage (in-lever length/out-lever length) (hereafter "MA") correlates strongly with PC1 scores ($R = 0.89$, $p < 0.001$; Supplementary Fig. S13). This pattern is consistent with other studies of jaw shape variation in vertebrates, in which MA appears consistently as one of the most important functional variables[30,56,57,59]. This reflects a trade-off between jaws with high bite force but slow closing versus jaws with low bite force but rapid closing. This trade-off reflects well-studied differences in prey-capture strategies[21,60,61]. Meanwhile, the dorsoventral depth of the symphysis correlates most strongly with PC2 scores ($R = 0.41$, $p < 0.001$; Supplementary Fig. S14). Taxa in the positive PC2 extreme have a deep symphysis, which is what would be expected of jaws that resist transverse torsional kinesis. These taxa are primarily dalatiid, oxynotid, and somniosid sharks which are characterised by specialised gouging dentitions in which there is only a single row of teeth within the bite, aligned to single

blade-like arcade[62–64]. Taxa on negative PC2 scores have anteroposteriorly broader dental grooves near the symphysis. They consist of triakids, hemigaleids, and heterodontids. These have broader, rasp-like dentitions near the symphysis, more consistent with grasping and prey manipulation[50,65,66].

There is some ambiguity in the above interpretations, however. Extracting values like torsional resistance and MA from shark mandibles is challenging, given there are few discrete landmarks on these structures. Using inter-landmark distances to calculate these values is, at best, a rough proxy. A better approach to mechanical advantage, for instance, would involve direct comparisons of adductor muscle diameters to lower jaw length. Future investigations could analyse how to shape summaries observed here co-vary with details of dental morphology more broadly to generate a fuller functional morphospace. However, in general terms, the results are consistent with patterns observed in other vertebrate groups, as noted above.

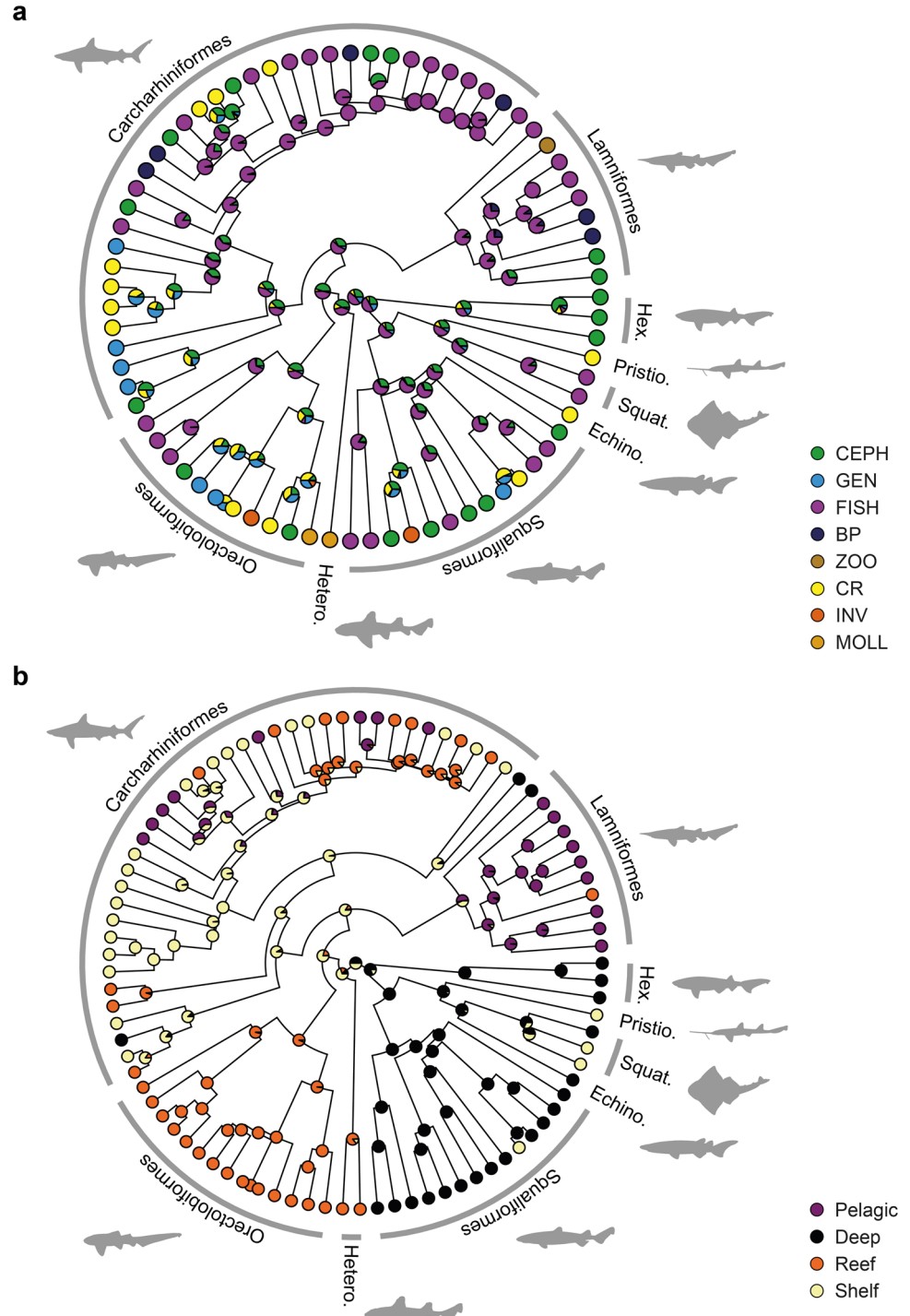

**Fig. 4 Ancestral state analysis of diet content and habitat for the species analysed. a** Diet content and **b** habitat. Pie charts at the nodes indicate the probabilities inferred for each trait. Tips colours indicate the state of the trait. Silhouettes based on outlines from Ebert et al. (2021). Diet abbreviations as in Fig. 2.

Consistent with the correlation between functional traits and the PCs, the correlations with ecological data support a relationship between mandible shape and macroecology. We show that habitat is a strong predictor of jaw shape (Table 1). However, if we account for body size, other factors like diet, trophic level, and taxonomic order (Table 1) also represent strong predictors of shape. We will discuss the relationship to taxonomic order in the section on phylogenetic patterns below. Trophic level and diet categories summarise the same data (diet contents) and thus

would be expected to correlate. Similarly, body size is a widely regarded proxy for trophic level[60]. Thus, the specifics of jaw morphology are important in addition to body size, which is significant in making inferences about ecology. This has important ramifications for palaeoecological studies. It validates the underlying assumptions of the relationship between morphological disparity and functional disparity through the time of early gnathostomes and the ecological diversity through jaw morphology and dental characteristics[29–31,67,68].

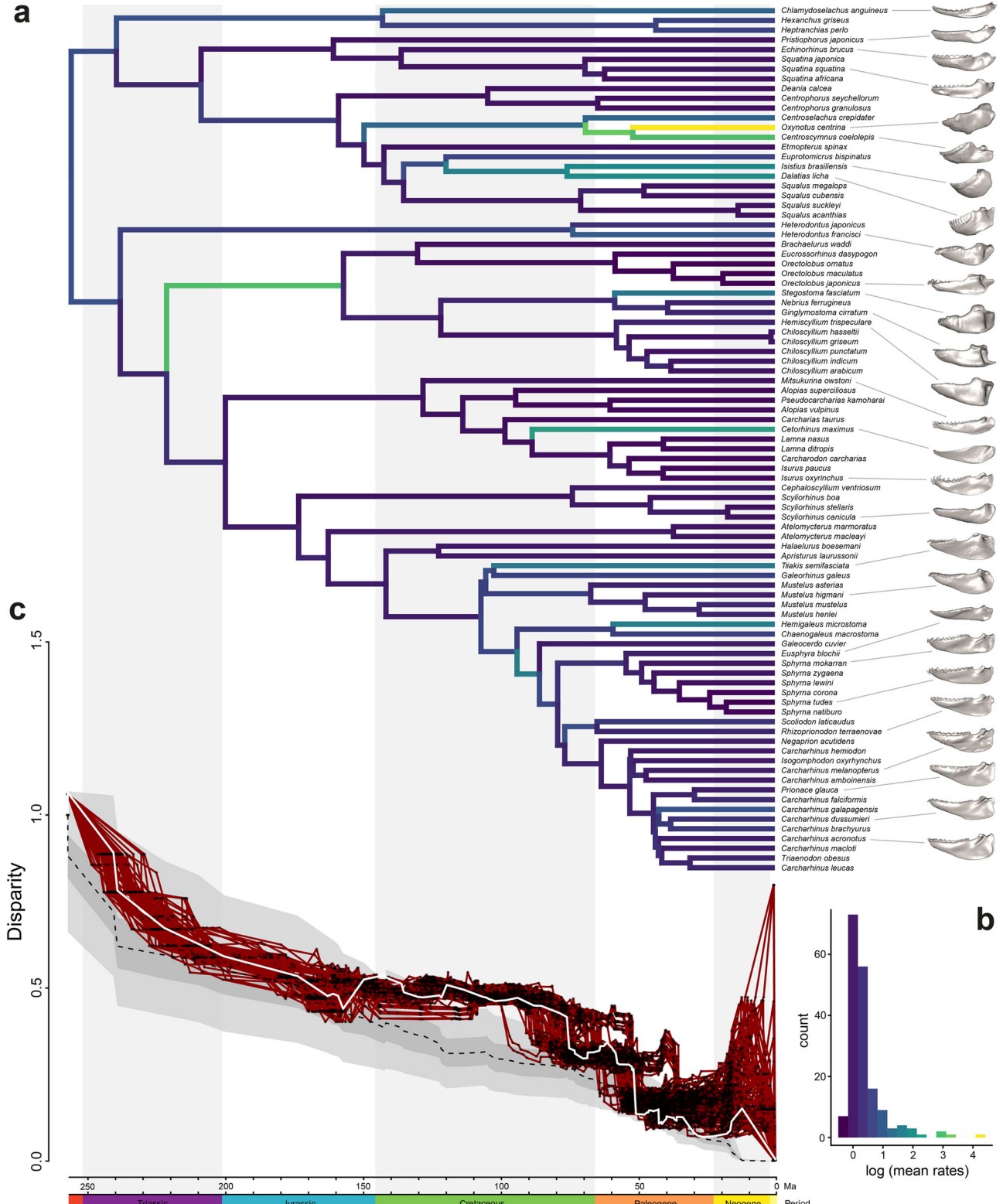

**Fig. 5 Evolutionary rates of jaw shape among modern sharks.** Shifts in jaw shape evolution by branch based on the variable rates model in BayesTraits (**a**) branches on the phylogeny indicate faster rates (warmer colours) and lower rates (colder colours) (**b**). Disparity through time plot for jaw shape (**c**). Solid white line: observed subclade disparity for the maximum credibility tree; dashed line: Brownian motion expectation; shaded area: 95% confidence interval of Brownian motion simulations. Solid red lines: observed subclade disparity for each tree from the random subsample (100 trees) and the relative node position indicated by the black dots. Representative jaws reconstructions for each order are indicated by a dashed line.

We identify two 'hotspots' of morphological diversification in shark mandibles: jaw disparity and rates of jaw shape evolution were highest both in the reef and deep-water environments. This broadly agrees with patterns of speciation rate and habitat in sharks[24] and disparity in teleost fishes[28]. Although Sorenson et al.[24]. found that speciation rates were apparently elevated in deep-water habitats, this result was not statistically significant. Nevertheless, Sorenson et al.[24]. showed that speciation rates were not uniquely elevated in nearshore habitats. Later Claes et al.[69]. and Straube et al.[70]. found that in deep sea squaliforms high speciation rates are associated with bioluminescence. Additionally, the diversification pattern for galeomorphs in reefs is supported by a rather recent diversification event in orectolobiforms[71], which also is supported by the fossil record of orectolobiforms[72]. Our results show both significant levels of morphological disparity and elevated rates of morphological evolution (Fig. 3). This compares with the results of Martinez et al.[28]. who found much higher morphological diversity in deep-sea teleosts as compared to shallow-water realms. These findings highlight the importance of deep-water settings as important sources of novel biodiversity both in terms of species number and morphological (and functional) novelty[73–75].

The breakdown of the 'onshore-offshore' diversification gradient in fishes is an intriguing pattern. Although the patterns of disparity and morphological evolutionary rate that we recovered are in agreement with previous results of environmentally driven speciation rates patterns in sharks from Sorenson et al.[24], we find that disparity and diversity are not necessarily coupled (Fig. 3). In deep-water habitats, diversification rate and rates of morphological evolution are coupled; the majority of deep-water occupation is by squaliforms, hexanchiforms, pristioforms, echinorhiniforms and scyliorhinids (catsharks). Except for the latter, all of these belong to the Squalomorphii clade and thus demonstrate that this clade dominates the deep-water realm[76]. Meanwhile, the reef-centred disparity is high, but this disparity is spread across two major clades: orectolobiforms (carpet sharks) and carcharhiniforms (requiem sharks). The latter, however, exhibit relatively low morphological disparity and amongst the lowest evolutionary rates. Among reef-dwelling taxa, it is the orectolobiforms that appear to contribute most to elevated rates of morphological evolution compared to the more rapidly diversifying carcharhiniforms.

The decoupling of diversity and disparity by environment supports the conjecture of Sorenson et al.[24]. regarding the cause of an environmentally 'flat' diversification gradient. They proposed that a key driver of this pattern was the early occupation of the deep marine realm by squalomorph sharks, as suggested by Klug and Kriwet[77]. Foote[78] noted that diversity and disparity tend to be coupled early in a clade's history when it is undergoing its initial radiation. Meanwhile, the decoupling of diversity and disparity occurs after significant turnover. This can be detected in the disparity through time analysis, which is consistent with the increase in diversity through the observed fossil record of sharks (from the Late Jurassic to Late Cretaceous)[79,80]. The stable occupation of deep water by squalomorphs is thus consistent with the expected pattern for an early occupation followed by niche stability[81–83].

We also find support for Sorenson et al.'s[24] inference that morphological specialisation played a key role in the squalomorph dominance in deep-water habitats. Sorenson et al.[24]. implicated morphological innovations such as luminescent organs (see also[70]). However, the highly unique jaw shapes of etmopterids (rough sharks), somniosids (sleeper sharks) and dalatiids (cookie-cutter sharks) indicate that wholesale anatomical divergence played a key role[63]. Our results show that these sharks dominate a large region of morphospace, largely not overlapped by other shark taxa.

The ancestral state reconstruction displays divergent evolutionary trends towards different prey compositions (Fig. 4). Nevertheless, a widespread and consistent tendency towards higher trophic level prey content (sensu Cortes[1]) could be inferred in carcharhiniforms and lamniforms, both independently evolved into big predators from an inferred piscivory state. Small-sized taxa like orectolobiforms tend to feed on lower trophic level prey, while a wider range of prey could be observed in squaliform sharks (Fig. 4a and Supplementary Fig. S9). This is supported by the pMANCOVA (Table 1) of shape and diet, as also observed by Cortes[1] and Pimiento et al.[84]. Modern sharks show a high morphological disparity at their origin before decreasing in disparity until recent times, implying an early burst evolutionary model. Such patterns have been associated with declines in evolutionary rates[85]. Moreover, simulations of trait evolution suggest that discrete trophic levels may mirror an early burst model[86].

The observed patterns inferred from our analyses might be influenced by using *Callorhinchus milli* as an outgroup, particularly because holocephalans are characterised by highly specialised jaw morphologies[87]. In view of such potential bias, it might be reasonable to also incorporate both extinct non-holocephalan and holocephalan chondrichthyans, since they display a wide variety of lower jaw morphologies[88–92]. Moreover, the inclusion of fossil forms is expected to improve the evolutionary model selection and consequently the outcomes of further analyses[93]. For instance, the wide variety of lower jaw and tooth morphologies displayed by extinct hybodontiform shark-like chondrichthyans (e.g.[94–97]), which are supposed to form the closest sister group to elasmobranchs[98], suggests that hybodontiforms might have been as ecologically diverse as are modern elasmobranchs, although their facies distribution indicates a rather stress-tolerant euryhaline ecology (e.g.[99]).

Our work adds to a growing body of evidence that fish diversification has a complex relationship with environmental gradients. The well-established onshore-offshore diversification rate pattern is largely derived from benthic invertebrates, which have richly sampled, high-fidelity fossil records. However, for benthic invertebrates, the offshore realm may represent a more homogenous habitat structure than for nektonic groups. The water column of the offshore realm is vast and partitioned by photic zones, oceanic currents, and temperature fluctuations[100,101]. This highlights the importance of motile, nektonic animals in understanding the relationship between habitats/environments, phenotype, life history, and speciation dynamics. Our results confirm that deep-sea habitats and reefs are important centres of morphological evolution in sharks. This has significance for theories of habitat-mediated diversification. Our work potentially corroborates the hypothesis of Martinez et al.[28]. by revealing a "dual morphological hotspot" seen in fishes, but larger sample sizes in future studies will be needed to verify the strength of apparent rate shifts. Such corroboration could be a major dividend in studies of habitat-mediated diversification. Sharks allow us to partially rule out potential alternative explanations that may otherwise be teleost-specific, thereby allowing more precise identification of important habitat factors. Sharks and teleost fishes are not closely related, and thus this repeated pattern can be divorced from teleost-specific explanations, such as genome expansion. The jaws of modern sharks also lack the high-performance suction mechanisms of teleosts, and thus the distinction cannot also be linked primarily to 'key innovations' in feeding, at least. The strong correspondence between habitat and jaw morphology, as well as the low overlap between deep-water and shallow-water jaw morphospace in sharks, further supports the view that foraging style is a key explanation for these divergencies. Niche partitioning in the deep-water realm is supported by the wide morphospace

occupation of deep-water sharks. The coupling of diversity and disparity in the deep sea is consistent with a pattern of niche stability in squalomorphs. Furthermore, this coupling is consistent with the hypothesis that squalomorphs occupied the deep sea realm early in their lineage history.

## Methods

**Morphometric data.** Our data set consists of lower jaw surface mesh reconstructions of 145 individuals belonging to 90 species, representing nine orders. These meshes are available as described in Kamminga et al.[102] and Dearden et al.[103]. For subsequent analyses, we defined two data subsets: one comprising only shark species ($n = 89$); and a second subset based on species with available stomach content data ($n = 75$). We defined a total of six landmarks, 51 curve sliding landmarks and 53 surface landmarks, to describe the three-dimensional shape of the jaws. All the landmarks and curve semilandmarks were taken by the same person (F.A. L.-R.) using the software Landmark Editor (Version 3.0)[104]. To capture the surface landmarks across all individuals, we first used a reference specimen as a template (*Carcharhinus acronotus*) onto which all the fixed, curve and surface landmarks coordinates are captured. We used the template to place the surface landmarks on the rest of the individuals in a semiautomated process with the Morpho R package[105] (Supplementary Information Methods and Supplementary Fig. S1).

**Phylogenetic tree.** For the phylogenetic relationships, we selected a topology reflecting resolved relationships at the order level based on the distribution of 1000 trees of the selected species, obtained from http://vertlife.org/sharktree/ [106]. From these trees, we generated a maximum credibility tree in TreeAnnotator version 1.8.2[107] and a subset of 100 trees to account for phylogenetic uncertainty. We used the subset of trees for further analyses with the geometric morphometric data (Available at https://github.com/Faviel-LR/Jaws_Sharks_Evolution).

**Trophic and ecology data.** To categorise the stomach content among all the selected species, we gathered information from the literature. We used mainly the data from Cortés[1] and complemented the information for several species not included in his study, as well as updating data for some of the species (Supplementary Data 1). With the prey categories defined by Cortés[1], we used the proportions of prey to estimate diet dissimilarity between species with the Bray-Curtis index. Afterwards, we clustered the categories with a UPGMA to define feeding guilds (Supplementary Data 2). Since some of the species present only one diet category, we calculated the clustering dendrogram a second time, excluding species with monospecific diets. We performed the dissimilarity and clustering analyses using the vegdist function in the R vegan package[108] and the hclust function in the R stats package[109]. With these results, we used the guilds as categorical variables to group the species as fish consumers (FISH; $n = 27$), cephalopod consumers (CEPH; $n = 18$), crustacean consumers (CR; $n = 12$), generalists (GEN; $n = 7$), larger prey consumers (e.g., marine mammals, other chondrichthyans, here referred to as big predators (BP; $n = 6$), molluscs consumers (excluding cephalopods) (MOLL; $n = 3$), invertebrate consumers (excluding molluscs, crustaceans and zooplankton) (INV; $n = 2$), and zooplankton consumers (ZOO; $n = 1$). We performed additional comparisons considering the trophic level value[110], with values lower than 3.8 as low-level predators (LP; $n = 20$), from 3.81 to 4.2 as mesopredators (MP; $n = 41$), and values greater than 4.2 as top predators (TP; $n = 29$). We considered the ecological lifestyle to investigate possible associations of morphological variation and habitat. Accordingly, we follow the habitat categories defined by Dulvy et al.[111], distinguishing reef, shelf, pelagic, and deep-sea distributions. In total, the species in our data set are reef ($n = 28$), shelf ($n = 24$), pelagic ($n = 17$) or deep-sea ($n = 21$) associated (see Supplementary Information Methods, Supplementary Data 3).

**Data analysis.** With the landmark coordinates (100 coordinates; 6 landmarks and 94 semilandmarks, Supplementary Figs. S1 and S2), we performed a generalised Procrustes alignment, in which the curves and surface semilandmarks are allowed to slide in order to minimise the bending energy and to avoid semilandmarks passing across an anatomical landmark[112]. To identify the major axes of shape variation, we performed a PCA with the Procruste's aligned coordinates. Likewise, a phylogenetic-aligned component analysis was performed on the species level. We conducted both analyses using the gm.prcom function in geomorph[113]. To visualise the landmark variation along each PC, we used the procrustes.var.plot in the landvR package[114]. We also quantified the phylogenetic signal under the Brownian motion model with $K_{mult}$[115], based on 1000 iterations with the landmark data averaged by species with the maximum credibility tree.

To investigate for shape differences between the group categories, we performed a phylogenetic multivariate analysis of variance type II (pMANOVA), using the averaged by species Procrustes aligned landmark coordinates. A pMACOVA of the shape and centroid size interaction suggests that size is significant but contributes little to the shape variation ($r^2 = 0.04085$; $F = 3.705$; $Z = 2.3226$; $p = 0.027$). Therefore, we did not correct for size. Likewise, to test for the effect of the categories on the shape data while accounting for size covariation (using the logarithm transformed centroid size), we performed a phylogenetic multivariate analysis of covariance type II (pMANCOVA). We fitted a multivariate phylogenetic

linear model with Pagel's lambda using penalised likelihood with the mvgls function in the mvMorph package[116]. We estimated the significance of each of the generated models with the manova.gls function[117] using Wilks Λ as a test statistic and 1000 permutations to account for differences in sample size.

We estimated the rates of morphological evolution for each category (order, diet content, trophic level, and habitat). First, we fitted the evolutionary model (equal rates, symmetric, all rates different) for each category with the 100 trees subsample using the fit_mk function in the castor package[118]. We selected the model based on the distribution of the Akaike information criterion (AIC) and the log-likelihood from all the fitted models (Supplementary Table S1). With the selected model for each category, we made a stochastic character mapping on the 100 trees subsample using the make.simmap function in the phytools package[119]. We used silhouettes from Ebert et al.[120]. to illustrate trait evolution. Afterwards, we used the character-mapped trees and the aligned landmark coordinates averaged by species with the function mvgls in the package mvMorph under a Brownian Motion model[116]. We extracted the model parameter estimates to obtain the evolutionary rates as described in Fabre et al.[121].

Next, we compared the morphological disparity for each category with the aligned landmarks using the dispRity.per.group function, we used the sum of variances as metric and 100 bootstraps, to overcome differences in sample size. Likewise, we estimated the disparity through time with the dtt.dispRity function. For this, we used the trimmed tree with only shark species and their respective averaged aligned landmarks; the sum of variances was used as a metric. Both analyses were performed with the dispRity package[122]. To estimate differences in morphological disparity for each group subset (order, habitat, trophic level, diet composition), we used the Wilcoxon test with Bonferroni correction with the test.dispRity function.

We applied a multivariate variable rates model in BayesTraits v 3.0.2 (http://www.evolution.rdg.ac.uk) to estimate the rates of morphological evolution under a Bayesian framework. We used as variables the principal components from the phylogenetically aligned PCA deemed important (first 4 PCs) based on a log-likelihood ratio[123], as implemented in the package Morpho with the function getMeaningfulPCs. Next, we implemented in BayesTraits the reversible-jump Markov Chain Monte Carlo (rjMCMC) algorithm to analyse the rate shifts for continuous trait evolution. We set five independent chains running for 200,000,000 iterations, with the first 25,000,000 discarded as burn-in and sampling was done every 20,000 iterations. Afterwards, we evaluated the chains' convergence by analysing the trace plots and the effective sample size (ESS > 200). To evaluate the chains' convergence, we used Gelman and Rubin's diagnostic in the coda package[124] (Supplementary Fig. S5, Supplementary Tables S2 and S3). With these results, we summarised the average branch-specific rate with the rjpp function in the BTRTools package (https://github.com/hferg/btrtools/tree/master/R) and plotted the rates on the maximum credibility tree with the plotBranchbyTrait function in phytools[119]. Additionally, we evaluated whether evolutionary rates display shifts in distinct regimes (i.e., habitat, trophic level, and diet content). We used a pool of 100 stochastic mappings with the make.simmap function in phytools to account for phylogenetic uncertainty. Next, we used a rjMCMC run over 20,000,000 generations with two independent runs and a burn-in of 10%. The convergence of the runs for each one of the regimes was assessed with Gelman and Rubin's diagnostic (Supplementary Table S4). This analysis was implemented in the package ratematrix[125]. Finally, we estimated the ancestral jaw shape considering all the averaged coordinates after GPA by species, for which we used the anc.recon function in Rphylopars[126].

**Reporting summary.** Further information on research design is available in the Nature Portfolio Reporting Summary linked to this article.

## Data availability

All data sources used in the study are indicated in the methods section. The ply files for the shark species can be found at https://doi.org/10.6084/m9.figshare.c.3662366.v1 and 10.18563/journal.m3.133 for *C. milii*. The tables, landmarks, and phylogenetic trees are deposited at https://github.com/Faviel-LR/Jaws_Sharks_Evolution.

## Code availability

The R script is available at https://github.com/Faviel-LR/Jaws_Sharks_Evolution.

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

## Acknowledgements

This research was funded in part by the Austrian Science Fund (FWF) [P33820]. For the purpose of open access, the author has applied a CC BY public copyright licence to any Author Accepted Manuscript version arising from this submission. F.A.L.R. is supported by the PhD Completion Grant from the Vienna Doctoral School of Ecology and Evolution. A.P. is supported by the ANR grant "ANR # ANR-22-CE02-0015-01_MACHER". We would like to thank Thomas Guillerme, and three anonymous reviewers whose comments helped to improve the manuscript.

## Author contributions

M.D.B, F.A.L.R. and J.K. conceived the project. F.A.L.R., M.D.B., S.S. and J.K. drafted the manuscript. S.S. and F.A.L.R. produced the figures. C.B. provided analytical tools and assistance. P.K. and A.P. performed CT data acquisition and reconstruction. F.A.L.R. performed the data analysis. All authors contributed equally to the interpretation of results. All authors revised the paper.

## Competing interests

The authors declare no competing interests.
