## [Peer Review File · Communications Biology]

Reviewers' comments:

Reviewer #1 (Remarks to the Author):

This manuscript describes results of an analysis of shark jaw diversification based on a landmark data set from a sample of 145 specimens from 90 species. An extensive series of results is reported but the emphasis is placed on the finding that rates of jaw shape evolution are highest in the deep sea. The results are placed in the context with an already substantial literature on depth and diversification in teleost fishes.

There are some things I like about this manuscript. I like that this team is developing large data sets on shark morphology. Both phylogenetic and morphological data on sharks (and other chondrichthyans) have lagged behind work on teleosts which is surprising as sharks are a much smaller and more manageable group. The question of whether sharks show high morphological diversification in the deep sea is reasonably interesting – as this is what has been recently shown for body shape evolution in teleosts. But this manuscript is very difficult for the reader. There are very substantial difficulties with the writing of the manuscript, the logic in many places, the methods and the interpretations. Below, I enumerate what I see as the chief problems with the paper and then I offer a long list of specific comments.

1. The core issue in the paper is whether sharks in different major habitat realms show different patterns of morphological diversity. This is translated into a habitat variable with four states. The problem is that the authors also create other discrete traits (diet and taxonomic order) and conduct all the same analyses on these traits. This makes for an exceedingly complicated set of results which leads to some deeply problematic graphics (e.g. Figure 3) and mismatches between text citations and figures/tables (e.g. see comments below about L231 and L234 where the wrong tables are cited). In essence, the presentation of the results is chaotic and does not focus on the main points.

2. The study is motivated by past work on speciation rates in sharks (Sorenson et al. 2014) and body shape evolution in teleosts (Martinez et al. 2021). But the results of the first paper are interpreted quite loosely – indeed I would say incorrectly. That paper found not significant overall effect of habitat on speciation rate in sharks. But this manuscript takes the trend in that study as demonstrating higher speciation rate in the deep. This paper also states that Sorenson found high speciation rates on reefs but that result only applied to one group of reef-swelling sharks, not reef dwelling sharks in general. By misrepresenting the Sorenson paper in this way, the current paper sets up a false 'conventional wisdom' about shark evolution to contrast their results with. This ms also never distinguishes between their focus on jaw shape evolution and the paper by Martinez et al 2021 who studied overall body shape evolution in teleosts. This tendency to be too loose in using past work and interpreting their own results makes for a rather frustrating manuscript.

3. The very extensive set of analyses that were done in this paper raise methodological concerns. My advice here is that the authors focus on the analyses that are germane to their central questions rather than showing every result they ever got in exploring their data set. A major example is that disparity and rates of evolution of jaw shape is calculated on the landmark coordinate data but comparisons of rates of jaw shape evolution are conducted on a hugely reduced representation of this data set – the first few PC axes. It's not clear that these are comparable and there are papers that show that doing rates analyses on PC scores biases results. Other analyses are misinterpreted (e.g. the DTT plot in Fig 4 is described incorrectly). In the end I did not have confidence in the results of the study.

4. Sample size is a problem in the study at many scales. There is an average of 1.6 specimens per species but the analyses are done on 'species averages' many of what are averages of one specimen. This will inflate rate estimates but there is no attempt to quantify the effect of such small sample sizes. Plots of the data (e.g. fig 1) show all the individuals without discriminating which are duplicates of individuals in the same species. Then the species are put into groups, many of which are too small for reliable estimates of disparity or rate. The result is that minimally, the results here need to be taken as very preliminary and awaiting more robust data sets.

5. The manuscript is not written clearly. The combination of misinterpreting past studies and only loosely conveying ideas makes for a manuscript where the purpose and conclusions are not as clear as they should be.

L60-61. This sentence is confusing. How are 'environments' different from 'habitats'? And what is meant by referring to 'genomics' as a driver of diversification? The sentence is vague and should be rewritten.

L68. What 'contrasts' are you referring to? This sentence is confusing because it appears to mark a difference in pattern in different groups but there is no explanation.

L73. What is 'display an extraordinary diversity...'? How much diversity is 'extraordinary'?

L90-91. 'potential competing drivers'? In what sense are the potential drivers competing? I suggest deleting the word 'competing' from this sentence.

L94-96. Sorry, I do not understand the logic here? Sharks lack teleost novelties so this 'allows a more refined set of explanatory parameters.'? This is unclear. What do you mean?

L98-99. What 'classic onshore-offshore diversification gradient' in marine invertebrates are you referring to? Citations?

L99-102. This is not a complete sentence and does not make sense as written. Give a citation for the diversification rate result. Why do say 'appear to have exhibited'? And why does the result on phenotypic evolution in teleosts complement a result on speciation rates in sharks? I am having trouble following the meaning and the logic. And, 'furthermore' seems an awkward term here?

L103. Which benthic invertebrates? There are lots of water-column invertebrates as well.

L104. Can you be more specific about 'morphological specializations in squalimorph sharks playing a role in the early occupation of the deep sea'?

L119. You mention filter feeding but are filter feeders included in this study?

L116. I think you should reverse the order of this sentence that starts at the end of line 116 and the sentence that starts middle of L121.

L125. Delete 'up to now'.

L128. What is meant here by 'following this'. In what sense does this work follow the work that characterizes shark diets? I don't understand.

L130. 'how morphological diversification in the feeding system evolved...' Morphological diversification is evolution. Diversification does not evolve, morphology does. The sentence needs to be rewritten.

L131. What do you mean that you 'provide crucial and novel information'. This is more hyperbole. 'crucial'?

L137. In figure 1, clarify in the caption whether the 'n = X' numbers are number of species or specimens.

L137. The sample sizes per species in this study are quite problematic and there is no attempt to address the error of estimating species values with such small samples. This is potentially a serious issue that inflates estimates of the Brownian rate parameter and needs to be justified quantitatively, not just with some hand-waving.

L137. It is also a problem for Figure 1 that the reader does not know which points are the same species and which are different species. For example, the two squaliforms in the upper right of the

plot. Is this a single species or two species? And how about the squaliforms that score high on PC2 but middle on PC1? Is that five species, four, three, two or one? The implications are substantial and the figure is very difficult to interpret when there are more than one point per species but the reader does not know what is what. This absolutely must be addressed.

L141. "We consider high PC1 scores to related to high mechanical advantage." This is a very problematic statement. High MA of what? In the case of muscle-skeleton system MA would refer to a specific action (e.g. biting, jaw opening), a specific location for the output force (where on the tooth row), a specific joint, and a specific muscle. So the reference here is vague and unclear. Also, if you want to discuss MA, why not measure it instead of asserting that PC1 is explained by MA. It is almost certainly true that even if PC1 is correlated with MA (and this should be demonstrated, not asserted), PC1 also probably has other elements of shape in it.

L145. "In PC2, positive values appear to relate to torsional resistance." The same problem as the previous comment. If you want to measure torsional resistance you should directly measure the morphological features that relate to it. There is clearly much more going on here than variation in torsional strength.

L145. Also, what is a 'gouging-type jaw'? You are asserting functional properties and functions without documentation or rationale. This should be avoided. Demonstrate the existence of 'gouging' jaw shape, do not just assert it?

L148. Again, you interpret the shape difference to indicate great adductor insertion area in specimens that have low PC2 score. But you did not actually measure area of adductor insertion and there is no analysis that shows us the relationship between PC2 and area of adductor insertion. The interpretation takes too much liberty with interpreting the PC axis.

L168. Is this a habitat effect or is it just different clades occupy different areas of morphospace. I am not convinced that these factors have been separated here.

L168. Regarding figure 2, the PaPCA appears to simply be a plotting of the phylogeny in the space of the original PCA – this is not a phylogenetic PCA as it seems to be described.

L177-186. But what about suspension feeders that eat plankton? It is a big problem that they are not included here as it seems clear they would have a large impact on disparity and rate.

L189-190. This sentence needs to be rewritten for clarity.

L210. Table 1. In the legend should it be 'centroid size' not 'cetroid size'? Also, why does the MANOVA have to do with centroid size – I do not understand this legend for the table.

L222. This statement about smaller species is not at all obvious from Figure S8D. It is unclear that there is any significant different in habitat distribution across body size. The numbers are simply too small and I do not see any quantitative analysis that supports this interpretation.

L205-223. I strongly recommend that this section be rewritten for clarity. It's unclear how the entries in table 1 related to these descriptions. It is also problematic that there are so many P-value reported in the table without any correction for multiple tests.

L225. I am confused about why you rely on disparity and rate analyses. In a Brownian motion model disparity accumulates in a clade linearly with time. The whole point of estimating the rate parameter is to remove the effects of time and phylogeny from standing disparity. So unless you want to make a specific point about the disparity it just seems strange to treat them as separate issues. This is like the relationship between species number, time and speciation rate. You would not meaningfully compare species richness between groups that differ 5-fold in age. You would compare speciation rate to remove the confounding effect of time.

L226. This first sentence needs to be rewritten for clarity. Disparity is a variance. Differences in variance among groups would not be interpreted as 'differences in jaw shape between orders'.

What they would show is differences in the diversity of jaw shape between groups.

L226. Also, the Y-axis scale in figure 3 is too compressed to see differences in rate. This is also because the rate estimates are so similar in every category.

L231. The table S3 that is cited here has nothing to do with comparison of disparity?

L234. The cited table S4 shows nothing about disparity in diet groups as stated here?

L228-240. In these comparisons you should give the disparity of the two groups. For example, in L228 PV for squaliforms is 0.036 but what is it for 'remaining orders'. And be clear about whether a difference is higher or lower. I am surprised that orectolobiforms differ from the remaining orders as they appear in a plot of PC1&2 to be similar in dispersion.

L231-240. I am concerned that the number of species in some of the diet categories is too small to generate meaningful estimates of disparity. This is acknowledged in Table S6 but still – are variances of groups with 5 species meaningful? I do not see how they could be. And the entries in Table S6 seem impossible. These significance tests are all significant at $P < 2.5 \times 10^{-33}$. That's 33 decimal places. I have never heard of P values that small?! This just does not make sense given that some of the disparity values are actually quite similar (Gen = 0.024, CR = 0.026: how could these actually be different?). Something is not right.

L241. Sentence needs to be rewritten. Maybe: "Comparisons between orders of the evolutionary rate of the lower jaw shape show that..."

L241-243. These rates for orectolobiforms, squaliforms and hexanchiforms are all very similar.

L257. Fig 3 is very problematic because the Y axis is too expanded to see the differences in any of the rate. Also, it is problematic to group and regroup the species in the study this way and estimate rate. I recommend that rates be reported in relative fashion. For example, the rate for deep-sea appears high but it is not even half again as high as the rate for reefs.

L262. This is another example of a sentence that needs to be carefully rewritten for clarity. I think you are saying that in all orders the diet category changes several times, but I'm not certain and I also doubt that is correct since some order have so few species in them.

L267. And here I think you are trying to point out cases of convergence in diet but the language is not at all clear.

L270. It is unclear how these estimates of rate in orders are different from the ones discussed on the previous page? If you have two ways here you should pick one, not report both. Figure 4 is a problem the way it is discussed here because there is no way to related this background rate variation to specific groups.

L284. This result for an early burst model is quite different from what is shown in the DTT plot in figure 4, which shows that jaw shape evolution never departs from Brownian motion. This does not support the idea of an early burst.

L288. I am completely baffled by this analysis (Table S10-12).

L330. I do not follow the logic that the fact you estimated ancestral jaw shape 'validates the underlying assumptions in recent studies...' I do not understand what you are saying.

L306-308. You actually have not shown that major axes of shape variation are consistent with functional interpretations – rather you have asserted that. There is no analysis in the paper of the relationship between jaw shape and mechanics or other aspects of function.

L308-309. You also have not shown that jaw shape is a reasonable proxy for feeding ecology (I think you mean diet components).

L309. This statement about PC1 and jaw mechanical advantage is an assertion. This point was never carefully addressed in the paper – you just asserted it.

L322. But why would habitat be a strong predictor of jaw shape? We expect jaw shape to be related to feeding mode. But a general habitat category like the ones used here ('deep', 'reef') does not seem like it should be a huge force. If prey are diverse in reefs I could see habitat resulting in higher rates, but why should it affect mean shape?

L337. "We identify two hotspots of morphological diversification in shark mandibles:..." This is an awkward use of the term 'hotspot'. Earlier in the paper a hotspot was a habitat where diversification has high. Here it is a metric of diversification. It would be best to use the term in only one way to avoid confusion. It is also problematic that disparity and rate of evolution are two features of the same thing, not really different aspects of diversity. Rate (the Brownian rate parameter) is calculated as a function of disparity and the phylogeny (essentially time). So these are not two different metrics of diversification.

L351. I guess I am not so clear that the results reported here are so consistent with Sorenson et al 2014. They did not find a significantly higher rate of speciation in the deep lineages. In one large clade they did find high speciation rate on reefs – carcharinids. Should that be taken as strong support for an 'onshore-offshore' gradient? I don't think so. Reefs only influenced speciation in one group, not the others that occupy reefs. That means it's not a general effect.

L355. Further, Sorenson et al. did not find that deep lineages have significantly higher speciation rate. I think it is playing a bit loose with their results to then conclude that high speciation rate and rates of jaw shape evolution are coupled in deep sharks.

L366-375. Here you state that diversity and disparity are decoupled but in the previous paragraph you said they are coupled? It cannot be both. Also I do not follow how consistent presence in the deep by squalomorphs supports the ideas of Foote? There is no analysis of niche stability here.

L376. In the introduction of the paper you stated that sharks do not show innovations but here you discuss bioluminescence as an innovation.

L385. It would be helpful if you could elaborate on this point about 'distinct feeding strategies'. What are the distinct, novel feeding strategies represented by this area of morphospace?

L388. This first sentence is very confusing. '...divergent evolutionary paths towards different prey compositions.' What do you mean? There are no analyses of 'evolutionary paths' such as sequences of transitions, pathways through morphospace, etc. So where does this come from?

L420. It is problematic that this 'onshore-offshore diversification pattern' that is apparently classic is never described in this paper and none of that literature is cited. What is that pattern? Please cite the main papers.

L429. I fail to see how 'nektonic' marine animals have been 'overlooked' as subjects in studies of diversification. There are a number of papers that describe the effect of ocean depth on fish communities, both at the scale of marine teleosts using very modern techniques (e.g. Martinez et al. 2021, Myers et al. 2021; Carrington et al. 2021). And depth is a well-known driver of speciation in marine fish (Gaither et al. 2018 Nature Ecology & Evolution).

L432. "This has significance for theories of habitat-mediated diversification." Could you explain what significance you have in mind here? Not clear.

L434. What is the dual morphological hotspot that Martinez et al 2015 identified that you mention here? Unclear.

L436. '...not especially closely related...' What does this mean? Be more specific.

L439. I do not follow the argument. In what way has anyone argued that high suction feeding performance is why teleosts diversify phenotypically in the deep sea?

L441. How is a pattern of 'diversity and disparity' consistent with the hypothesis that squalomorphs occupied the deep sea early in their history? It is not clear what one has to do with the other?

L522. Were the MANOVAs run on landmark data or the PC scores (as described later)?

L542. This description suggests that the estimates of rate were done in separate analyses from the test of differences in rate. This seems problematic.

L555. I am concerned that the landmark coordinates were not used in these analyses, but instead the first few PC axes. There is a substantial literature that shows how problematic it is to study evolutionary dynamics of multivariate data sets by focusing on the major PC axes (see paper by Josef Uyeda).

L572. Using multiple stochastic maps does not account for phylogenetic uncertainty as stated here – it accounts for uncertainty in the history of the discrete trait.

Reviewer #2 (Remarks to the Author):

*** Editor's note: the format for our Articles does not permit Methods before the Discussion ***

In this study, López-Romero et al. study the evolution of shark mandibles as a means to evaluate the environmental factors driving morphological diversification. Specifically, they test the 'onshore-offshore' diversification gradient hypothesis, whereby shallow waters in the continental shelf represent a hotspot for diversification. They find that shark morphological evolution peaks in both reefs and deep-water habitats, with deep-water sharks showing a higher disparity. I found this study very exciting and interesting. The findings seem highly relevant for marine biology and evolution with high potential to influence future research. However, I found some mostly editorial or structural issues which prevented me from providing a deep evaluation of this work. Essentially, I found inconsistencies between Fig. 3 and the text, and found Fig. 3 specially hard to read. Consequently, I was unable to judge some of the results interpretations as I would have liked to. In addition, I think the paper would benefit from either bringing the Methods before the Results if allowed by the journal, or if not, to guide the reader with a brief overview of the methods before starting describing the results. Similarly, for clarity, it may be a good idea to merge the Results and the Discussion. Finally, in terms of the science, the paper will benefit from a discussion on the fact that today, the highest diversity of sharks takes place in the continental shelf and not in deep waters, which brings the question: if the deep-ocean played such an important role in the evolution of sharks, how/why did the continental shelf become a hotspot of biodiversity today?

Below I provide more feedback per section.

Intro:

Very well-written and clear.

Results:

-Provide a short summary of the methods to guide the reader. Otherwise, it is rather a shock to go from the intro to the results section with very technical language, some of it may be meaningless to the wider readership in my opinion.

-Some terminology is unclear. For example, what is mechanical advantage?

-The TL abbreviation is often used to denote total length. I suggest to use "troph" or other.

-227-2459: There seems to be a mix-up here with the figures. 3A is "mean evolutionary rate" but it is cited when referring to disparity and vice-versa. There are other inconsistencies here: for example, I don't see how Orectolobiformes has the highest mean evolutionary rate in the figure (the highest seems to be Squaliformes, red box in Fig 3B) or that the CEPH group evolved faster than the rest of the groups.

-Some results are presented as a finding, but they are well-known aspects of shark biology. For example, L210-212: "lamniforms and carcharhiniforms are among the largest species, while squaliforms and orectolobiforms are the smallest" or -L264: "the majority of species have a piscivorous diet".

-L263-269: What figure or table can the reader use to follow these results? Fig S9 seems to be about diet and not size.

Discussion:

-Overall, well-written and clear, however, hard to given the inconsistencies mentioned above.

-When referring to trophic ecology, the findings are discussed along with body size (e.g., L391: carcharhiniforms and lamniforms, both independently evolved into big predators from an inferred piscivory state). However, at this point, it remains unclear how body size was factored in here.

Methods:

-How did you deal with sampling differences amongst orders?

-L528: Space missing after point.

-Many shark species feed on multiple food items, how was this accounted for in the stomach content analyses?

Figures:

-Figure 3A: Perhaps a log scale would facilitate the interpretation. Fig c-f is disproportionately large when considering its relevance. I suggest moving it to supplement and improve a-b as they are hard to read.

-Supplementary figures are poorly labeled and have poor quality: Fig S8: hard to see what the colours denote.

-Figs S8-S9: Please explain abbreviations.

-I think fig. S9 should be in the main text

Reviewer #3 (Remarks to the Author):

López-Romero et al. assess the evolution of shark jaw morphology in the context of the 'onshore-offshore' diversification gradient, thus shedding light on the environmental controls of species diversity. The authors find support for the hypothesis that sharks constitute a striking exception to the general pattern, demonstrating that high rates of morphological evolution occur not only in reef but also in deep-water habitats.

I congratulate the authors for this interesting piece of work, and especially for tackling this broad-interest question with such a detailed methodological procedure. I think the methodology is robust (but see some comments below) and the conclusions are sound and well supported. Therefore, I strongly support the eventual publication of this article in *Communications Biology*. I have some comments that the authors might want to consider in reviewing their manuscript. I feel they could have run some extra 'sensitivity' analyses considering uncertainty associated to node calibrations, lifestyle categorizations, and some morphological outliers.

Lines 124-128. Isotopic approaches constitute good alternatives to stomach content analysis.

Among many other papers see:

Speed, C. W., et al. "Trophic ecology of reef sharks determined using stable isotopes and telemetry." *Coral reefs* 31.2 (2012): 357-367.

Hussey, Nigel E., et al. "Stable isotope profiles of large marine predators: viable indicators of trophic position, diet, and movement in sharks?." *Canadian Journal of Fisheries and Aquatic Sciences* 68.12 (2011): 2029-2045.

MacNeil, M. Aaron, Gregory B. Skomal, and Aaron T. Fisk. "Stable isotopes from multiple tissues reveal diet switching in sharks." *Marine Ecology Progress Series* 302 (2005): 199-206.

Estrada, James A., et al. "Predicting trophic position in sharks of the north-west Atlantic Ocean using stable isotope analysis." *Journal of the Marine Biological Association of the United Kingdom* 83.6 (2003): 1347-1350.

Speed, C. W., et al. "Trophic ecology of reef sharks determined using stable isotopes and telemetry." *Coral reefs* 31.2 (2012): 357-367.

McCormack, Jeremy, et al. "Trophic position of *Otodus megalodon* and great white sharks through

time revealed by zinc isotopes." *Nature Communications* 13.1 (2022): 1-10.
Kast, Emma R., et al. "Cenozoic megatooth sharks occupied extremely high trophic positions." *Science Advances* 8.25 (2022): eabl6529.

Lines 157-160. Most squaliforms also cluster in a comparatively narrow area of the morphospace (PC1-2, and PC1-3), and only a few taxa (I think Somniosidae and Oxynotidae) occupy most extreme positive values in PC1. Could the results be driven by those "outliers"? Did the authors consider rerunning (at least some) analyses excluding this taxa?

Lines 351-365. The discussion on the whether or not diversification rate and rates of morphological evolution is confusing and I think it could be better expressed. The authors did not perform any analyses of (taxonomic) diversification rates, how then they tested this? From data in Sorenson et al. (2014)? If so, a figure showing taxonomic diversification rates and morphological change rates would be very welcome (even in the supplementary material).

In addition, the fact that Carchariniiformes in 'reefs' display low disparity and morphological evolutionary rates does not entail that diversity and disparity are decoupled in the different environments. Actually, reefs and deep-waters show high speciation rates (see Sorenson et al. 2014) and high morphological disparity. The difference is that in the former only orectolobiformes contribute to this pattern (but it is still true) while, in the second case, apparently all the clades do.

Lines 387-417. As it is, it seems these analyses/results are not contextualized within the general rationale of the paper. I wonder whether the authors could better explain why providing a reconstruction of the ancestral jaw morphology and diets is important for the main goal of this article (i.e., testing hypotheses of diversification gradients and their linking to morphological specialization).

In this sense, the authors recognise that the inclusion of *Callorhynchus milli* could have biased their ACSR analyses. I wonder whether they could expand on why the inclusion of an outgroup is important here and why not to include other potentially more suitable (extinct) taxa (closest to the ancestral node of elasmobranch).

Line 477. Do the set of 100 trees consider node-calibration uncertainty? This constitute another potential source of uncertainty that should be accounted for.

Lines 485 and 499. Tables S1 and S2 do not contain this information. Do the authors mean Data S1 and S2?

Lines 487-488. How did the authors determine the number of clusters for establishing dietary groups? The authors might better specify which are the criteria they followed for this.

Lines 504-507. I found the terminology used for categorizing lifestyles/habitats confusing. Actually, several of the species categorized as pelagic inhabit mostly the continental shelf. I would say the "shelf" species here constitute actually demersal (or benthopelagic) sharks. A neritic species ('shelf' species) could be either demersal or pelagic; and a pelagic species could be either neritic or oceanic.

Lines 504-507. Did the authors consider alternative categorization of the species in order to account for potential uncertainty in their classification in lifestyles? This would be advisable as the classification of some species is certainly non trivial.

Lines 510-512. The employment of minimum bending energy to slide semilandmarks is not arbitrary and should be justified in the main text/supplementary methods.

Please revise the citations to supplementary files in the Supplementary Information Methods. I found several inconsistencies.

Reply to reviewers' comments

Reviewer #1 (Remarks to the Author):

This manuscript describes results of an analysis of shark jaw diversification based on a landmark data set from a sample of 145 specimens from 90 species. An extensive series of results is reported but the emphasis is placed on the finding that rates of jaw shape evolution are highest in the deep sea. The results are placed in the context with an already substantial literature on depth and diversification in teleost fishes.

There are some things I like about this manuscript. I like that this team is developing large data sets on shark morphology. Both phylogenetic and morphological data on sharks (and other chondrichthyans) have lagged behind work on teleosts which is surprising as sharks are a much smaller and more manageable group. The question of whether sharks show high morphological diversification in the deep sea is reasonably interesting – as this is what has been recently shown for body shape evolution in teleosts. But this manuscript is very difficult for the reader. There are very substantial difficulties with the writing of the manuscript, the logic in many places, the methods and the interpretations. Below, I enumerate what I see as the chief problems with the paper and then I offer a long list of specific comments.

1. The core issue in the paper is whether sharks in different major habitat realms show different patterns of morphological diversity. This is translated into a habitat variable with four states. The problem is that the authors also create other discrete traits (diet and taxonomic order) and conduct all the same analyses on these traits. This makes for an exceedingly complicated set of results which leads to some deeply problematic graphics (e.g. Figure 3) and mismatches between text citations and figures/tables (e.g. see comments below about L231 and L234 where the wrong tables are cited). In essence, the presentation of the results is chaotic and does not focus on the main points.

- *The fundamental point of the paper is to understand whether lower jaw geometry is useful as an ecological proxy, as is commonly assumed in e.g. palaeobiological studies. Habitat realms predict, inter alia, availability of different prey phenotypes, different locomotor needs and thus different foraging styles. The degree to which jaw shape predicts diet is key as this is very much the 'first-order' assumption palaeobiologists are trying to make. We have restructured the introduction to make it clear that the paper does intentionally have a broad focus directed at a broad audience. Additionally, we addressed the problems mentioned with the figures and tables, as it has been suggested by the other reviewers as well.*

2. The study is motivated by past work on speciation rates in sharks (Sorenson et al. 2014) and body shape evolution in teleosts (Martinez et al. 2021). But the results of the first paper are interpreted quite loosely – indeed I would say incorrectly. That paper found not significant overall effect of habitat on speciation rate in sharks. But this manuscript takes the trend in that study as demonstrating higher speciation rate in the deep. This paper also states

that Sorenson found high speciation rates on reefs but that result only applied to one group of reef-swelling sharks, not reef dwelling sharks in general. By misrepresenting the Sorenson paper in this way, the current paper sets up a false 'conventional wisdom' about shark evolution to contrast their results with. This ms also never distinguishes between their focus on jaw shape evolution and the paper by Martinez et al 2021 who studied overall body shape evolution in teleosts. This tendency to be too loose in using past work and interpreting their own results makes for a rather frustrating manuscript.

- *As we will detail below, the Reviewer's characterisation of our interpretations of Sorenson et al. are not correct. We never interpreted Sorenson et al. as showing particularly elevated speciation rates in deep water. We did point out that they recovered such a result and we explicitly noted that they found that this was not significant. What we do point to is the fact that they failed to recover evidence of an environmental gradient in speciation rates (the null hypothesis). But that is the relevant point itself, as an onshore-offshore diversification gradient in sharks would predict a significant relationship. We were using the term "elevated" relative to the expectations of onshore-offshore patterns in which there is a significantly lower diversification rate in the deep than in shallow marine environments. We admit that we should have been more precise in our wording, something corrected in the present draft.*

3. The very extensive set of analyses that were done in this paper raise methodological concerns. My advice here is that the authors focus on the analyses that are germane to their central questions rather than showing every result they ever got in exploring their data set. A major example is that disparity and rates of evolution of jaw shape is calculated on the landmark coordinate data but comparisons of rates of jaw shape evolution are conducted on a hugely reduced representation of this data set – the first few PC axes. It's not clear that these are comparable and there are papers that show that doing rates analyses on PC scores biases results. Other analyses are misinterpreted (e.g. the DTT plot in Fig 4 is described incorrectly). In the end I did not have confidence in the results of the study.

- *We explain in the reply to the specific comment the rationale for using a reduced set of PC scores. Besides, this was done for a single analysis with the ratematrix package. The rest of the analyses were done using the full set of landmarks. We strongly disagree with the observation regarding the DTT plot in Figure 4. We further provide simulations of other evolutionary models and their impact on the pattern observed in the disparity results.*

4. Sample size is a problem in the study at many scales. There is an average of 1.6 specimens per species but the analyses are done on 'species averages' many of what are averages of one specimen. This will inflate rate estimates but there is no attempt to quantify the effect of such small sample sizes. Plots of the data (e.g. fig 1) show all the individuals without discriminating which are duplicates of individuals in the same species. Then the species are put into groups, many of which are too small for reliable estimates of disparity or rate.

The result is that minimally, the results here need to be taken as very preliminary and awaiting more robust data sets.

- *We do agree with the reviewer in this point. In many cases the acquisition of individuals is also limited by their rarity in collections. Surely the inclusion of many more specimens per species will improve the inferences drawn from the results. At the same time, despite this limitation, our study opens the doors to further research on variation within sharks' species. Regarding the species averages, this was only performed for the purpose of the phylomorphospaces. The rest of the comparisons were done with larger groups (Orders, Habitat, Trophic Level, Diet Groups) except for the comparisons between orders, since we were interested in Hexanchiformes and Squatiniformes rates. We further addressed this comment with questions from Reviewer 3, where we subject our results to different metrics (in particular of Disparity) and removing outliers from the data.*

5. The manuscript is not written clearly. The combination of misinterpreting past studies and only loosely conveying ideas makes for a manuscript where the purpose and conclusions are not as clear as they should be.

- *In addition to Reviewer 2's comments, we have sought to refine the manuscript to give a better 'high-level' view of the total set of analyses and objectives.*

L60-61. This sentence is confusing. How are 'environments' different from 'habitats'? And what is meant by referring to 'genomics' as a driver of diversification? The sentence is vague and should be rewritten.

- *"Environment" frequently refers to a set of abiotic conditions, while "habitat" refers to the assemblage of organisms living together in a locale as well as the abiotic environment. As indicated in the references cited, it is suggested that genome duplication specific to teleosts might have an impact on their diversification. Redaction changed, now reads: "Key targets of this research are the relative roles of environment, genomics, and novelty..."*

L68. What 'contrasts' are you referring to? This sentence is confusing because it appears to mark a difference in pattern in different groups but there is no explanation.

- *Here, we are referring to onshore-offshore and latitudinal biodiversity gradients (as already stated in the first draft version of the manuscript) : "striking contrasts in large-scale biodiversity gradients (such as onshore-offshore diversification, origination of higher taxa preferably on nearshore environments, later expanding offshore in their evolutionary history (Jablonsky et al, 1983) and latitudinal biodiversity gradients)". We rephrased now in Lines 91-94: "(such as onshore-offshore diversification, origination of higher taxa preferably on nearshore environments, later expanding offshore in their evolutionary history (Jablonsky et al, 1983) and latitudinal biodiversity gradients)".*

L73. What is ‘display an extraordinary diversity...’? How much diversity is ‘extraordinary’?

- *With about 35,000 species, teleosts account for half of all extant vertebrates. That seems to be a remarkable number in comparison with sharks. However, we rephrased this sentence to sound less hyperbolic. Now reads in lines 77 & 78: “with nearly 35,000 living species (Fricke et al, 2022), teleosts represent a vast taxonomic inventory”*

L90-91. ‘potential competing drivers’? In what sense are the potential drivers competing? I suggest deleting the word ‘competing’ from this sentence.

- *Done.*

L94-96. Sorry, I do not understand the logic here? Sharks lack teleost novelties so this ‘allows a more refined set of explanatory parameters.’? This is unclear. What do you mean?

- *As the sentence reads, the novelties present in teleosts- have been suggested to drive their diversification. This is the case of acanthomorphs (which comprise around 85% of bony fish’ diversity). We changed the redaction, now in lines 83 & 84 of the revised manuscript: “Sharks also lack some of the specific novelties of the feeding system and genome that have been attributed to the species richness of teleosts”*

L98-99. What ‘classic onshore-offshore diversification gradient’ in marine invertebrates are you referring to? Citations?

- *We mentioned this in lines 89-95 of the revised manuscript with their corresponding references. See comment above.*

L99-102. This is not a complete sentence and does not make sense as written. Give a citation for the diversification rate result. Why do say ‘appear to have exhibited’? And why does the result on phenotypic evolution in teleosts complement a result on speciation rates in sharks? I am having trouble following the meaning and the logic. And, ‘furthermore’ seems an awkward term here?

- *We have changed the redaction in lines 96-98 of the revised manuscript: “Sorenson et al. (2014), for instance, found equivocal results for an onshore-offshore diversification gradient in sharks. More pointedly, Martinez et al. (2021), identified the deep-water realm as a crucial hotspot for morphological diversity of bony fishes.”*

L103. Which benthic invertebrates? There are lots of water-column invertebrates as well.

- *We have changed the redaction, now this sentence is excluded from the text.*

L104. Can you be more specific about ‘morphological specializations in squalimorph sharks playing a role in the early occupation of the deep sea’?

- *Specifically, Sorenson et al. (2014) mention bioluminescence. We elaborate in the discussion (lines 410-414).*

L119. You mention filter feeding but are filter feeders included in this study?

- *The dataset includes the basking shark *Cetorhinus maximus*, (which was the only one available).*

L116. I think you should reverse the order of this sentence that starts at the end of line 116 and the sentence that starts middle of L121.

- *We prefer to keep the sentence order as it is.*

L125. Delete ‘up to now’.

- *Done*

L128. What is meant here by ‘following this’. In what sense does this work follow the work that characterizes shark diets? I don’t understand.

- *Redaction changed we removed “following this” and start at line 103 of the current version “The purpose of this paper is two-fold”*

L130. ‘how morphological diversification in the feeding system evolved...’ Morphological diversification is evolution. Diversification does not evolve, morphology does. The sentence needs to be rewritten.

- *Here, we are referring to disparity. We have changed the sentence accordingly. Now in line 125: “to explore how morphological disparity in the feeding system evolved”.*

L131. What do you mean that you ‘provide crucial and novel information’. This is more hyperbole. ‘crucial’?

- Changed to: "...we provide novel information about potential biotic and abiotic drivers..." Line 126 of the revised manuscript.

L137. In figure 1, clarify in the caption whether the 'n = X' numbers are number of species or specimens.

- Done.

L137. The sample sizes per species in this study are quite problematic and there is no attempt to address the error of estimating species values with such small samples. This is potentially a serious issue that inflates estimates of the Brownian rate parameter and needs to be justified quantitatively, not just with some hand-waving.

- *We are aware of the limitations of using species-level comparisons in this study, and the variation at even genus level, which is also dependent on the structure analysed. For instance, fin shape variation seems to be important for batoids within species. This is the reason we selected comparisons between groups with larger samples. To further corroborate the possible variation, we estimated the shape variation between orders and genera with a Procrustes ANOVA. The shape variation within each group indicates that at the genus level there is less variation than within the order. This is evident by their F values: Order SS = 2.374, $R^2 = 0.46339$, $F = 59.029$; Genus SS = 2.019, $R^2 = 0.45636$, $F = 12.168$. As a side note, even at different shape scale analyses, it has been shown that more recently diverging groups, at species level, show higher variation between species than within (see Gunstra et al., 2021).*

*Grunstra, N. D., Bartsch, S. J., Le Maître, A., & Mitteroecker, P. (2021). Detecting phylogenetic signal and adaptation in papionin cranial shape by decomposing variation at different spatial scales. *Systematic Biology*, 70(4), 694-706.*

L137. It is also a problem for Figure 1 that the reader does not know which points are the same species and which are different species. For example, the two squaloforms in the upper right of the plot. Is this a single species or two species? And how about the squaloforms that score high on PC2 but middle on PC1? Is that five species, four, three, two or one? The implications are substantial and the figure is very difficult to interpret when there are more than one point per species but the reader does not know what is what. This absolutely must be addressed.

- *Species labels can be found in Suppl. Fig. 4. Unfortunately, given the severe overlap of many points, it is almost impossible for the program to perform the task without losing information.*

L141. “We consider high PC1 scores to related to high mechanical advantage.” This is a very problematic statement. High MA of what? In the case of muscle-skeleton system MA would refer to a specific action (e.g. biting, jaw opening), a specific location for the output force (where on the tooth row), a specific joint, and a specific muscle. So the reference here is vague and unclear.

- *We have modified the text to make it clear we are referring to jaw-closing mechanical advantage. Implicitly, this is over the tooth row because we use the front and back of the tooth row as landmarks for the out- and in-levers respectively.*

Also, if you want to discuss MA, why not measure it instead of asserting that PC1 is explained by MA. It is almost certainly true that even if PC1 is correlated with MA (and this should be demonstrated, not asserted), PC1 also probably has other elements of shape in it.

- *We have now measured it using the landmarks for the front and rear of the tooth row against the rear of the articular as a means of estimating out- and in-lever dimensions. This is far from perfect, but it allows us to correlate estimated jaw-closing MA with PC scores. With these ratios, we plotted the results against PC1 (for MA) and PC2 (for Torsional Resistance (TR), addressing the comment below), we found both values are correlated with the PC scores ($R = 0.89$, $p < 0.001$ for PC1 and MA; $R = 0.41$, $p < 0.001$ for PC2 and TR). We include this information now in the discussion and provide the supplementary figures (Figures S12 & 13). It is indeed certainly the case that there are other shape contributors to PC1, but there are also almost certainly other correlates of jaw-closing MA that go beyond in-/out-lever ratios (similarly, MA almost certainly also contributes to other PC axes). This does not mean we cannot explore the apparent functional significance of shape variation along different axes. This is fairly common practice (see references below).*

*Anderson, P. S., Friedman, M., & Ruta, M. (2013). Late to the table: diversification of tetrapod mandibular biomechanics lagged behind the evolution of terrestriality. *Integrative and Comparative Biology*, 53(2), 197-208.*

*Morales-García, N. M., Gill, P. G., Janis, C. M., & Rayfield, E. J. (2021). Jaw shape and mechanical advantage are indicative of diet in Mesozoic mammals. *Communications biology*, 4(1), 242.*

*Johnson, M. M., Foffa, D., Young, M. T., & Brusatte, S. L. (2022). The ecological diversification and evolution of Teleosauroidea (Crocodylomorpha, Thalattosuchia), with insights into their mandibular biomechanics. *Ecology and Evolution*, 12(11), e9484.*

L145. “In PC2, positive values appear to relate to torsional resistance.” The same problem as the previous comment. If you want to measure torsional

resistance you should directly measure the morphological features that relate to it. There is clearly much more going on here than variation in torsional strength.

- *Same as above.*

L145. Also, what is a ‘gouging-type jaw’? You are asserting functional properties and functions without documentation or rationale. This should be avoided. Demonstrate the existence of ‘gouging’ jaw shape, do not just assert it?

- *This has already been documented several times since the summary presented by Moss (1977): “... The relatively long jaws of lamniform and carcharhiniform sharks, the rotation of the jaw apparatus and the efficient cutting dentition produce a feeding mechanism that is unusual in the predatory world; that is, the ability to efficiently carve chunks out of a prey that is too large to be entirely surrounded by the open mouth. With the evolution of this gouging feeding mechanism the relative safety afforded a prey organism by sheer size or bulk alone became less important.”*

Moss, S. (1977). Feeding mechanisms in sharks. American Zoologist, 17(2), 355-364.

L148. Again, you interpret the shape difference to indicate great adductor insertion area in specimens that have low PC2 score. But you did not actually measure area of adductor insertion and there is no analysis that shows us the relationship between PC2 and area of adductor insertion. The interpretation takes too much liberty with interpreting the PC axis.

- *We refer here to the surface on lateral view of the posterior part of the Meckel’s cartilage. Unfortunately, a term for this region has not been yet defined, only in selected species which display a flange on the most posterior margin (named as sustentaculum), nevertheless, this region corresponds to where the adductor fossa is located. We refer to it in the text as articular/adductor insertion region.*

L168. Is this a habitat effect or is it just different clades occupy different areas of morphospace. I am not convinced that these factors have been separated here.

- *We addressed this in Table 1 with the Phylogenetic MANOVA and MANCOVA. In both cases the phylogenetic signal (as estimated with Pagel’s lambda) indicate that there is a strong signal. This is more evident now when we observe the trait map on the tree in Fig S9 (Now Figure 4b). Some orders like Orectolobiformes and Squaliformes display a consistent distribution on a specific habitat.*

L168. Regarding figure 2, the PaPCA appears to simply be a plotting of the phylogeny in the space of the original PCA – this is not a phylogenetic PCA as it seems to be described.

- *This PaPCA aligns the data to the phylogenetic signal. Please see Collyer & Adams (2021): “...when data have no phylogenetic signal, PACA does not offer an appreciably different outcome than Phy-PCA”.*

*Collyer, M. L., & Adams, D. C. (2021). Phylogenetically aligned component analysis. *Methods in Ecology and Evolution*, 12(2).*

- *Here we show that the data not aligned to the phylogenetic signal, differs from the presented in figure 2, not only on the orientation of the data, but also regarding how much of the variation is explained by the components.*

On the left we present the original phylogenetically aligned PCA, on the right the phylomorphospace.

L177-186. But what about suspension feeders that eat plankton? It is a big problem that they are not included here as it seems clear they would have a large impact on disparity and rate.

- *As already mentioned earlier, there are a couple of *Cetorhinus maximus* specimens. In the case of both disparity and morphological evolutionary rates, it does not seem to change radically the values obtained for Lamniformes, which at the end present some of the lowest values for both measurements. Indeed, it would be great to include other filter feeding species like *Rhynchodon typus* and *Megachasma pelagios*, but unfortunately they are not yet available for study.*

L189-190. This sentence needs to be rewritten for clarity.

- *Redaction changed: “The prey content groups display a considerable overlap in the phylomorphospace”*

L210. Table 1. In the legend should it be 'centroid size' not 'cetroid size'? Also, why does the MANOVA have to do with centroid size – I do not understand this legend for the table.

- *Typo corrected. We are using centroid size as a covariate, therefore it is a MANCOVA. We have changed the legend accordingly.*

L222. This statement about smaller species is not at all obvious from Figure S8D. It is unclear that there is any significant difference in habitat distribution across body size. The numbers are simply too small and I do not see any quantitative analysis that supports this interpretation.

- *Since the regression score of the MANCOVA with centroid size as covariate indicate differences, we repeated this analysis with the `procDlm` function in `geomorph`, which was used to produce the plots presented in Suppl. Fig. 8. We have included a supplementary table (Table S10) to show all the comparisons made.*

L205-223. I strongly recommend that this section be rewritten for clarity. It's unclear how the entries in table 1 related to these descriptions. It is also problematic that there are so many P-value reported in the table without any correction for multiple tests.

- *MANOVA/MANCOVA does not requires a correction for multiple tests. This is usual when there are multiple pairwise comparisons in a post-hoc analysis, which is not the case. We do not see a problem with this table's legend. If any, we just introduce the values of Wilks Lambda as the F-Statistic associated on the table legend.*

L225. I am confused about why you rely on disparity and rate analyses. In a Brownian motion model disparity accumulates in a clade linearly with time. The whole point of estimating the rate parameter is to remove the effects of time and phylogeny from standing disparity. So unless you want to make a specific point about the disparity it just seems strange to treat them as separate issues. This is like the relationship between species number, time and speciation rate. You would not meaningfully compare species richness between groups that differ 5-fold in age. You would compare speciation rate to remove the confounding effect of time.

- *Overall, the groups we compared can be traced back to around the same time period (Whitenack et al, 2022). If disparity has accumulated linearly through time, modern shark orders would be expected to display similar levels of disparity. We show here that this is not the case, and in particular the most speciose orders show the decreased disparity in comparison to the other orders. Furthermore, when we did comparisons of disparity differences considering alternative grouping and removing outliers (as suggested by Reviewer 3) we found our initial results to be robust enough. We further*

elaborate that the pattern of disparity through time suggests an Early Burst model, in the comment for L284.

Whitenack, L. B., Kim, S. L., & Sibert, E. C. (2022). Bridging the gap between chondrichthyan paleobiology and biology. In Biology of Sharks and Their Relatives (pp. 1-29). CRC Press.

L226. This first sentence needs to be rewritten for clarity. Disparity is a variance. Differences in variance among groups would not be interpreted as ‘differences in jaw shape between orders’. What they would show is differences in the diversity of jaw shape between groups.

We used the Procrustes variance to measure it. We just changed the redaction to “The results of the morphological disparity, as Procrustes variance (PV), indicate significant differences in the jaw shape diversity between orders”.

L226. Also, the Y-axis scale in figure 3 is too compressed to see differences in rate. This is also because the rate estimates are so similar in every category.

- *From one of the estimations among the 100 trees, one of the results for Orectolobiformes was this extreme value, this distorted the plot. To correct this, we have adjusted the Y-axis with a break to allow for a better observation of the results’ dispersion.*

L231. The table S3 that is cited here has nothing to do with comparison of disparity?

- *Corrected, it is Suppl. Tab. 5*

L234. The cited table S4 shows nothing about disparity in diet groups as stated here?

- *Same as above.*

L228-240. In these comparisons you should give the disparity of the two groups. For example, in L228 PV for squaliforms is 0.036 but what is it for ‘remaining orders’. And be clear about whether a difference is higher or lower. I am surprised that orectolobiforms differ from the remaining orders as they appear in a plot of PC1&2 to be similar in dispersion.

- *The disparity is included for each order when introduced in the sentence and in the corresponding supplementary table. And perhaps the dispersion is similar, but the morphospace occupation is different. We illustrate this also in Suppl. Fig. 7, where the PC3 shows a bigger dispersion for Squaliformes. We explicitly mentioned this in lines 153-156 of the original text (now lines 146-152).*

L231-240. I am concerned that the number of species in some of the diet categories is too small to generate meaningful estimates of disparity. This is acknowledged in Table S6 but still – are variances of groups with 5 species meaningful? I do not see how they could be. And the entries in Table S6 seem impossible. These significance tests are all significant at $P < 2.5 \times 10^{-33}$. That's 33 decimal places. I have never heard of P values that small?! This just does not make sense given that some of the disparity values are actually quite similar (Gen = 0.024, CR = 0.026: how could these actually be different?). Something is not right.

- *Such small p values are not usually reported, probably because most people usually only report whatever their alpha level is. Also, in some cases the packages and software indicate only the printed value after a certain threshold (the procD.lm from geomorph for instance). Even base functions in stats from R like Kruskal.test can return a p value $< 2.2e-16$. Regarding the sample size, that is why the presented results are bootstrapped, which is a common procedure for such instances. Of course, a comparison of samples with only one individual would be meaningless for the purpose of variance, like in the case of Echinorhiniformes and Pristiphoriformes, for which we only had one species available.*
- *When considering the disparity, there can be different patterns of groups with low disparity and large species number, high disparity and high species number, low disparity and low species number and high disparity and low species number (Guillerme et al, 2020). In a specific case among sharks, we can see this pattern in groups like angel sharks (López-Romero et al, 2020).*

López-Romero, F. A., Stumpf, S., Pfaff, C., Marramà, G., Johanson, Z., & Kriwet, J. (2020). Evolutionary trends of the conserved neurocranium shape in angel sharks (Squatiniiformes, Elasmobranchii). Scientific reports, 10(1), 1-13.

L241. Sentence needs to be rewritten. Maybe: “Comparisons between orders of the evolutionary rate of the lower jaw shape show that...”

- *Sentence rewritten, now reads: “The comparisons of the mean evolutionary rate of the lower jaw between orders shows that...” in line 221 of the current manuscript.*

L241-243. These rates for orectolobiforms, squaliforms and hexanchiforms are all very similar.

- *That is the reason why we did the rates per branches analysis as shown in Fig. 4 (now Fig. 5). Besides the, the new figure shows the values more clearly.*

L257. Fig 3 is very problematic because the Y axis is too expanded to see the differences in any of the rate. Also, it is problematic to group and regroup the

species in the study this way and estimate rate. I recommend that rates be reported in relative fashion. For example, the rate for deep-sea appears high but it is not even half again as high as the rate for reefs.

- *Our intention was to show that deep sea species have comparable morphological evolutionary rates and disparity. We would like to keep the rate values in the original format.*

L262. This is another example of a sentence that needs to be carefully rewritten for clarity. I think you are saying that in all orders the diet category changes several times, but I'm not certain and I also doubt that is correct since some order have so few species in them.

- *Following the recommendation by Reviewer #2, we have moved Suppl. Fig. 9 to the main text in order to illustrate the results better.*

L267. And here I think you are trying to point out cases of convergence in diet but the language is not at all clear.

- *The sentence is not confusing, we prefer to keep it as it is. Besides we now include the ancestral state figure to support the paragraph.*

L270. It is unclear how these estimates of rate in orders are different from the ones discussed on the previous page? If you have two ways here you should pick one, not report both. Figure 4 is a problem the way it is discussed here because there is not way to related this background rate variation to specific groups.

- *The figure 3 show Evolutionary rates by groups and in figure 4 (now figure 5) indicate instances at specific branches, unlike the one presented in figure 3 in which we were mostly focussing on different groups which are not monophyletic.*

L284. This result for an early burst model is quite different from what is shown in the DTT plot in figure 4, which shows that jaw shape evolution never departs from Brownian motion. This does not support the idea of an early burst.

- *The result shown with the dtt function display the expected disparity under Brownian motion (Harmon et al. 2003). Surely, plenty of interpretations of these plots have been documented, but for a comparison of the continuous trait evolution under different evolutionary models Liam Revell shows how the dtt plots vary according to those models and how they are related to the biological process. The reviewer mentioned in a previous comment that in the BM model the disparity is accumulated linearly, and indeed that is the pattern shown here in the figure modelled after BM. But what we observe from our*

data is an EB model. Here are some plots of the simulated trait evolution and their observed disparity:

In this sense, the EB model indicated that the disparity accumulated early in the clade history, which is what we show in figure 4 (now 5). This pattern differs from the observed one under BM model. For further details into the functions please follow: <http://blog.phytools.org/2015/12/dtt-and-pattern-of-comparative-data.html>

L288. I am completely baffled by this analysis (Table S10-12).

- *As explained in the method section (lines 603-610 of the current version), this analysis is intended to explore if the variables (in this case the first two PCs) evolve in response to the different regimes (Diet, Trophic Level and Habitat).*

L330. I do not follow the logic that the fact you estimated ancestral jaw shape ‘validates the underlying assumptions in recent studies...’ I do not understand what you are saying.

- *Redaction changed, now reads: “Underlying assumptions of the relationship between morphological disparity and functional disparity through time of early gnathostomes”. Along with the references on the following lines of the paragraph.*

L306-308. You actually have not shown that major axes of shape variation are consistent with functional interpretations – rather you have asserted that. There is no analysis in the paper of the relationship between jaw shape and mechanics or other aspects of function.

- *Simply because we used qualitative assessments does not mean that the interpretations are bald-faced assertions. We based these interpretations on the relative warps plots. Our approach is commonly applied in morphometric studies: examine the relative warps diagrams for the extremes of each PC axis to explore which landmark positions appear to differ most strongly at each extreme (see e.g., Navalón et al, 2019; Morales-García et al, 2021;*

Galvez-López & Cox, 2022; Johnson et al, 2022). Nevertheless, to assuage the reviewer's concerns, we have provided plots of relevant morphological variables related to specific functional properties against PC scores to support our interpretations. The interpretations we have provided are fairly conservative in nature and consistent with the prior literature on jaw shape and function in fishes generally. We addressed this point in comment for line 141. We do not disagree that future quantitative studies could further evaluate those interpretations.

- *Navalón, G., Bright, J. A., Marugán-Lobón, J., & Rayfield, E. J. (2019). The evolutionary relationship among beak shape, mechanical advantage, and feeding ecology in modern birds. *Evolution*, 73(3), 422-435.*
- *Gálvez-López, E., & Cox, P. G. (2022). Mandible shape variation and feeding biomechanics in minks. *Scientific Reports*, 12(1), 4997.*
- *Johnson, M. M., Foffa, D., Young, M. T., & Brusatte, S. L. (2022). The ecological diversification and evolution of Teleosauroida (Crocodylomorpha, Thalattosuchia), with insights into their mandibular biomechanics. *Ecology and Evolution*, 12(11), e9484.*

L308-309. You also have not shown that jaw shape is a reasonable proxy for feeding ecology (I think you mean diet components).

- *We have tested both diet components and trophic level. This was the purpose also of the pMANOVA and pMANCOVA presented in Table 1. Additionally, we have indicated the value of Pagel's Lambda for those analyses, which also suggest a phylogenetic signal.*

L309. This statement about PC1 and jaw mechanical advantage is an assertion. This point was never carefully addressed in the paper – you just asserted it.

- *This was already pointed out by the reviewer in lines L 141, 145, 148, and 306-308, and we have rebutted this above. We have generated plots of approximate in-lever/out-lever ratios to show that this ratio correlates extremely strongly with the values of PC1, as is expected. We include now the description of the measurements in the supplementary information's methods.*

L322. But why would habitat be a strong predictor of jaw shape? We expect jaw shape to be related to feeding mode. But a general habitat category like the ones used here ('deep', 'reef') does not seem like it should be a huge force. If prey are diverse in reefs I could see habitat resulting in higher rates, but why should it affect mean shape?

- *Habitat is a predictor of jaw shape variation, not jaw shape. Different habitats can be expected to dictate different sets of feeding ecologies that reflect available prey types and foraging modes. See for instance Miller et al (2022), they show that some of the traits (e.g., jaw length) observed in bony fishes were important during the colonization of the deep sea, since it is a depauperate environment for food. Alternatively, as suggested by Reviewer #3, we explored other possible categories of habitat, and we observe the*

same pattern for the species distributing overall in deep waters. We elaborate further this question in the following comment.

Miller, E. C., Martinez, C. M., Friedman, S. T., Wainwright, P. C., Price, S. A., & Tornabene, L. (2022). Alternating regimes of shallow and deep-sea diversification explain a species-richness paradox in marine fishes. *Proceedings of the National Academy of Sciences*, 119(43), e2123544119.

L337. “We identify two hotspots of morphological diversification in shark mandibles:...” This is an awkward use of the term ‘hotspot’. Earlier in the paper a hotspot was a habitat where diversification has high. Here it is a metric of diversification. It would be best to use the term in only one way to avoid confusion. It is also problematic that disparity and rate of evolution are two features of the same thing, not really different aspects of diversity. Rate (the Brownian rate parameter) is calculated as a function of disparity and the phylogeny (essentially time). So these are not two different metrics of diversification.

- *We followed the term as presented by Martinez et al (2021), however, we agree that it might cause confusion since the term is usually applied to diversity hotspots. Both rates and disparity are not always linked, in fact it is proposed that some extreme morphologies can arise because of patterns of phenotypic integration (Felice et al, 2018). Although the possible relationship of the jaw morphologies and integration might be interesting to test here, several issues regarding the estimation of phenotypic integration (see: Cardini, 2020; Cardini & Marco, 2022; Felice et al, 2018; Mitteroecker & Schaefer, 2022; Zelditch & Goswami, 2021), can obscure the interpretation. Only recently other approaches have been proposed for datasets like ours with high number of variables (Conaway & Adams, 2022).*

*Cardini, A. (2020). Less tautology, more biology? A comment on “high-density” morphometrics. *Zoomorphology*, 139(4), 513-529.*

*Cardini, A., & Marco, V. A. (2022). Procrustes Shape Cannot be Analyzed, Interpreted or Visualized one Landmark at a Time. *Evolutionary Biology*, 1-16.*

*Conaway, M. A., & Adams, D. C. An effect size for comparing the strength of morphological integration across studies. *Evolution*.*

*Felice, R. N., Randau, M., & Goswami, A. (2018). A fly in a tube: macroevolutionary expectations for integrated phenotypes. *Evolution*, 72(12), 2580-2594.*

*Mitteroecker, P., & Schaefer, K. (2022). Thirty years of geometric morphometrics: Achievements, challenges, and the ongoing quest for biological meaningfulness. *American Journal of Biological Anthropology*.*

*Zelditch, M. L., & Goswami, A. (2021). What does modularity mean?. *Evolution & Development*, 23(5), 377-403.*

L351. I guess I am not so clear that the results reported here are so consistent with Sorenson et al 2014. They did not find a significantly higher rate of speciation in the deep lineages. In one large clade they did find high speciation rate on reefs – carcharinids. Should that be taken as strong support for an ‘onshore-offshore’ gradient? I don’t think so. Reefs only influenced speciation in one group, not the others that occupy reefs. That means it’s not a general effect.

- *This comment is difficult to address because it does not accurately reflect the wording of our paragraph and it is self-contradictory. It appears to be based on a reading of the cited paragraph without regards to the entire context. We never stated that Sorenson et al. found evidence of an onshore-offshore gradient; quite the opposite. The previous paragraph states explicitly “Although Sorenson et al. (2014) found that speciation rates were apparently elevated in deep-water habitats, this result was not statistically significant”. Secondly, our point is rather the opposite of what is stated by the reviewer: indeed, the lack of significant result supports a “breakdown of the ‘onshore-offshore’ diversification gradient”, using a verbiage quite similar to Sorenson et al’s own who repeatedly describe a “breakdown” of the onshore-offshore pattern.*

In further support of our interpretations, we have added a couple of references (line 371 of the current version) which actually show that deep sea Squaliformes display higher diversification rates in families which bear bioluminescence than the other families. Regarding the reefs, it is in particular for the Orectolobiformes which have experienced a recent radiation in reefs (Corrigan and Beheregaray, 2009) which is supported by the fossil record (Boyd and Seitz, 2021) and our analyses.

Straube, N., Li, C., Claes, J. M., Corrigan, S., & Naylor, G. J. (2015).

Molecular phylogeny of Squaliformes and first occurrence of bioluminescence in sharks. BMC evolutionary biology, 15(1), 1-10.

Claes, J. M., Nilsson, D. E., Mallefet, J., & Straube, N. (2015). The presence of lateral photophores correlates with increased speciation in deep-sea bioluminescent sharks. Royal Society open science, 2(7), 150219.

Boyd, B. M., & Seitz, J. C. (2021). Global shifts in species richness have shaped carpet shark evolution. BMC Ecology and Evolution, 21(1), 1-10.

Corrigan, S., & Beheregaray, L. B. (2009). A recent shark radiation: molecular phylogeny, biogeography and speciation of wobbegong sharks (family: Orectolobidae). Molecular Phylogenetics and Evolution, 52(1), 205-216.

L355. Further, Sorenson et al. did not find that deep lineages have significantly higher speciation rate. I think it is playing a bit loose with their results to then conclude that high speciation rate and rates of jaw shape evolution are coupled in deep sharks.

- *Same as above. We never said that they did say that; we explicitly acknowledged as much in the immediately preceding paragraph- and as we replied in line 337 comment - the pattern can be explored by analysing phenotypic integration, which is an ongoing project. The argument for coupling is not predicated on the results of Sorenson et al.'s work. The body of the paragraph actually explains how we arrive at the argument for coupling/decoupling of speciation and morphological disparity. The argument is fundamentally that in the deep sea, disparity is invested in the most speciose groups while in shallow carbonate settings, disparity is not invested in the most speciose groups. There is no implication in that paragraph that the argument is predicated on the results of Sorenson et al.*

L366-375. Here you state that diversity and disparity are decoupled but in the previous paragraph you said they are coupled? It cannot be both.

Again, this is not at all an accurate reflection of what our manuscript says. It is also a false assertion. It is possible for these metrics to be coupled in some groups while not coupled in others (in which sense, it can be both). As we noted, the metrics are coupled in deep water taxa, not coupled in shallow carbonate platform-dwelling taxa. To avoid confusion, we have included these references.

*Erwin, D. H. (2007). Disparity: morphological pattern and developmental context. *Palaeontology*, 50(1), 57-73.*

*Guillerme, T., Cooper, N., Brusatte, S. L., Davis, K. E., Jackson, A. L., Gerber, S., ... & Donoghue, P. C. (2020). Disparities in the analysis of morphological disparity. *Biology letters*, 16(7), 20200199.*

Also I do not follow how consistent presence in the deep by squalomorphs supports the ideas of Foote? There is no analysis of niche stability here.

The manuscript never claims to evaluate the ideas of Foote. However, the fact that diversity and disparity are coupled in squalomorphs is consistent with the predictions of niche stability. This paragraph is simply putting our research into context, which is the purpose of a Discussion section.

L376. In the introduction of the paper you stated that sharks do not show innovations but here you discuss bioluminescence as an innovation.

- *In the introduction we stated in lines 94-96 that sharks lack specific innovations seen in teleost fishes, not that sharks lack innovations. Sorenson et al. discussed bioluminescence as an innovation, We already addressed this in the comment for line 351.*

L385. It would be helpful if you could elaborate on this point about 'distinct feeding strategies'. What are the distinct, novel feeding strategies represented by this area of morphospace?

- *The study indicates different patterns in lower jaw length, with deep sea species displaying larger jaws. We have adjusted the paragraph accordingly. Lines 411-415 of the current manuscript: “However, the highly unique jaw shapes of etmopterids (rough sharks), somniosids (sleeper sharks) and dalatiids (cookie-cutter sharks) indicate that wholesale anatomical divergence played a key role (Underwood et al, 2016). Our results show that these sharks dominate a large region of morphospace, largely not overlapped by other shark taxa.”*

L388. This first sentence is very confusing. ‘...divergent evolutionary paths towards different prey compositions.’ What do you mean? There are no analyses of ‘evolutionary paths’ such as sequences of transitions, pathways through morphospace, etc. So where does this come from?

- *We changed the word “path” for “trend”.*

L420. It is problematic that this ‘onshore-offshore diversification pattern’ that is apparently classic is never described in this paper and none of that literature is cited. What is that pattern? Please cite the main papers.

In the introduction (l. 67-72) we mentioned the pattern, along with citations (Frédérich et al, 2016; Rabosky et al, 2018; Sallan et al, 2018), we just introduce briefly the description of the onshore-offshore pattern and the “origination of higher taxa preferably on nearshore environments, later expanding offshore in their evolutionary history”

*Jablonski, D., Sepkoski Jr, J. J., Bottjer, D. J., & Sheehan, P. M. (1983). Onshore-offshore patterns in the evolution of Phanerozoic shelf communities. *Science*, 222(4628), 1123-1125.*

*Frédérich, B., Marrama, G., Carnevale, G., & Santini, F. (2016). Non-reef environments impact the diversification of extant jacks, remoras and allies (Carangoidei, Percomorpha). *Proceedings of the Royal Society B: Biological Sciences*, 283(1842), 20161556.*

*Rabosky, D. L., Chang, J., Title, P. O., Cowman, P. F., Sallan, L., Friedman, M., ... & Alfaro, M. E. (2018). An inverse latitudinal gradient in speciation rate for marine fishes. *Nature*, 559(7714), 392-395.*

*Sallan, L., Friedman, M., Sansom, R. S., Bird, C. M., & Sansom, I. J. (2018). The nearshore cradle of early vertebrate diversification. *Science*, 362(6413), 460-464.*

L429. I fail to see how ‘nektonic’ marine animals have been ‘overlooked’ as subjects in studies of diversification. There are a number of papers that describe the effect of ocean depth on fish communities, both at the scale of marine teleosts using very modern techniques (e.g. Martinez et al. 2021, Myers et al. 2021; Carrington et al. 2021). And depth is a well-known driver of speciation in marine fish (Gaither et al. 2018 *Nature Ecology & Evolution*).

- *This assertion makes the point of what we mentioned. Most of the literature mentioned by the reviewer is recent. We changed the sentence to: “This highlights the importance of motile, nektonic animals in understanding the relationship between habitats/environments, phenotype, life history, and speciation dynamics.”*
- *And regarding the paper of Gaither et al. (2018), which is very interesting, in the first paragraph it states: “...While longitudinal and latitudinal habitat transitions have been proposed to define marine communities and promote intraspecific differentiation, little is known about the importance of transitions along ocean depth gradients.”*

L432. “This has significance for theories of habitat-mediated diversification.” Could you explain what significance you have in mind here? Not clear.

- *Besides the ones we already mentioned in the introduction (onshore-offshore and latitudinal gradients) there is also niche stabilization for instance.*

L434. What is the dual morphological hotspot that Martinez et al 2015 identified that you mention here? Unclear.

- *Mistake here, it is Martinez et al, 2021.*

L436. ‘...not especially closely related...’ What does this mean? Be more specific.

- *Redaction changed: “are not closely related” in line 469 of the current version.*

L439. I do not follow the argument. In what way has anyone argued that high suction feeding performance is why teleosts diversify phenotypically in the deep sea?

- *Is not related to diversification of teleosts in deep sea. But it has been indeed argued that suction feeding played an important role in bony fish diversification (Wainwright et al, 2015; Wainwright and Longo, 2017). This feeding mode is also presented among several shark species, and most notably among their sister group (Batoids), which differs from the one presented among teleosts (Wilga et al, 2007).*

*Wainwright, P. C., & Longo, S. J. (2017). Functional innovations and the conquest of the oceans by acanthomorph fishes. *Current Biology*, 27(11), R550-R557.*

*Wainwright, P. C., McGee, M. D., Longo, S. J., & Patricia Hernandez, L. (2015). Origins, innovations, and diversification of suction feeding in vertebrates. *Integrative and Comparative Biology*, 55(1), 134-145.*

Wilga, C. D., Motta, P. J., & Sanford, C. P. (2007). Evolution and ecology of feeding in elasmobranchs. Integrative and Comparative Biology, 47(1), 55-69.

L441. How is a pattern of ‘diversity and disparity’ consistent with the hypothesis that squalomorphs occupied the deep sea early in their history? It is not clear what one has to do with the other?

We provided references for this statement in the first version of the manuscript in Lines 401,402 and 408-415.

L522. Were the MANOVAs run on landmark data or the PC scores (as described later)?

- *Whole landmark data for both the pMANOVA and pMANCOVA.*

L542. This description suggests that the estimates of rate were done in separate analyses from the test of differences in rate. This seems problematic.

- *Because first of all, a discrete trait should be fitted, and after testing for the best supported model for discrete traits, these mapped traits on the trees are then used with the continuous variable. It is described step by step in the same paragraph.*

L555. I am concerned that the landmark coordinates were not used in these analyses, but instead the first few PC axes. There is a substantial literature that shows how problematic it is to study evolutionary dynamics of multivariate data sets by focusing on the major PC axes (see paper by Josef Uyeda).

- *Surely the comparisons with landmarks would include the total variation. Overall, it would be extremely difficult to implement these approaches with the whole landmark data. If we use the total number of variables (3X100 coordinates) it becomes utterly impractical. That is why we selected the principal components regarded as meaningful (see Bookstein, 2014, pp. 324). Besides, as pointed out recently, it is not recommended to analyse the results of disparity or rates per landmark level (Cardini, 2022).*

Bookstein, F. L. (2014). Measuring and reasoning: Numerical inference in the sciences. Cambridge University Press.

L572. Using multiple stochastic maps does not account for phylogenetic uncertainty as stated here – it accounts for uncertainty in the history of the discrete trait.

- *This is why we integrated our results over 100 tree topologies drawn from the posterior distribution, which does provide an accounting for phylogenetic uncertainty. The one presented at the end is only the Maximum Clade Credibility tree. It is mentioned in the previous lines “First, we fitted the evolutionary model (equal rates, symmetric, all rates different) for each category with the 100 trees subsample using the fit_mk function in the castor package (Louca and Doebeli, 2018).” Additionally we changed figure 4 to show the results of disparity with the 100 random trees, the dispersion of the DTT on the 100 trees shows differences on the nodes.*

Reviewer #2 (Remarks to the Author):

*** Editor's note: the format for our Articles does not permit Methods before the Discussion ***

In this study, López-Romero et al. study the evolution of shark mandibles as a means to evaluate the environmental factors driving morphological diversification. Specifically, they test the onshore-offshore' diversification gradient hypothesis, whereby shallow waters in the continental shelf represent a hotspot for diversification. They find that shark morphological evolution peaks in both reefs and deep-water habitats, with deep-water sharks showing a higher disparity. I found this study very exciting and interesting. The findings seem highly relevant for marine biology and evolution with high potential to influence future research. However, I found some mostly editorial or structural issues which prevented me from providing a deep evaluation of this work. Essentially, I found inconsistencies between Fig. 3 and the text, and found Fig. 3 specially hard to read. Consequently, I was unable to judge some of the results interpretations as I would have liked to. In addition, I think the paper would benefit from either bringing the Methods before the Results if allowed by the journal, or if not, to guide the reader with a brief overview of the methods before starting describing the results. Similarly, for clarity, it may be a good idea to merge the Results and the Discussion. Finally, in terms of the science, the paper will benefit from a discussion on the fact that today, the highest diversity of sharks takes place in the continental shelf and not in deep waters, which brings the question: if the deep-ocean played such an important role in the evolution of sharks, how/why did the continental shelf become a hotspot of biodiversity today? Below I provide more feedback per section.

Intro:

Very well-written and clear.

- *Thank you!*

Results:

-Provide a short summary of the methods to guide the reader. Otherwise, it is rather a shock to go from the intro to the results section with very technical language, some of it may be meaningless to the wider readership in my opinion.

As the editor mention, it is not possible to move the Methods section.

-Some terminology is unclear. For example, what is mechanical advantage?

- *Redaction changed; in the discussion section we explain the term “The jaw-closing mechanical advantage (in-level length / out-level length) (hereafter “MA”)” (line 316)*

-The TL abbreviation is often used to denote total length. I suggest to use “troph” or other.

- *We have changed the abbreviation to TR for consistency.*

-227-2459: There seems to be a mix-up here with the figures. 3A is “mean evolutionary rate” but it is cited when referring to disparity and vice-versa. There are other inconsistencies here: for example, I don’t see how Orectolobiformes has the highest mean evolutionary rate in the figure (the highest seems to be Squaliformes, red box in Fig 3B) or that the CEPH group evolved faster than the rest of the groups.

- *We have corrected the figure, now it expands the size of the boxplots, and we place the Figure 3 c-f to the supplementary information. In the case of Orectolobiformes, it is the outlier result on the top of the figure which makes the difference.*

-Some results are presented as a finding, but they are well-known aspects of shark biology. For example, L210-212: “lamniforms and carcharhiniforms are among the largest species, while squaliforms and orectolobiforms are the smallest” or -L264: “the majority of species have a piscivorous diet”.

- *Redaction changed here. We acknowledge that these observations were expected as reported in previous studies cited “This is supported by the pMANCOVA (Table 1) of shape and diet, as also observed by Cortes (1999) and Pimiento et al. (2019).”*

-L263-269: What figure or table can the reader use to follow these results? Fig S9 seems to be about diet and not size.

- *Specifically, for the regression of shape on size we presented the Fig S8, which is now corrected for the quality and labels. We include a table on the supplementary information with these results (Table S10).*

Discussion:

-Overall, well-written and clear, however, hard to given the inconsistencies mentioned above.

- *Thank you, we address the specific comments.*

-When referring to trophic ecology, the findings are discussed along with body size (e.g., L391: carcharhiniforms and lamniforms, both independently evolved into big predators from an inferred piscivory state). However, at this point, it remains unclear how body size was factored in here.

- *The results from the Table 1 indicate that when size is not included in the MANOVA, then there is not a significant effect. Additionally, we ran these analyses with the residuals of the shape regressed on the centroid size (Supplementary table 10), which indicate that with the regressed coordinates, there is a significant effect on the shape and Habitat but also with Diet (considering the 8 categories). A phylogenetic MACOVA of the shape and centroid size interaction suggest that size is significant but contributes little to the shape variation ($r^2 = 0.04085$; $F = 3.705$; $Z = 2.3226$; $p = 0.027$). Therefore, we did not correct for size using the residuals of the shape regressed on the centroid size. We have included this information in the supplementary information regarding the methods.*

Methods:

-How did you deal with sampling differences amongst orders?

- *In this case we selected only the orders which contained at least 3 species. This is an extreme case, since for some orders like Echinorhiniformes, or Pristiophoriformes, we only had one specimen available, and in the case of Heterodontiformes we only had two species available. We also address this pointed out by Reviewer #1 in line 137.*

-L528: Space missing after point.

- *Corrected, thank you!*

-Many shark species feed on multiple food items, how was this accounted for in the stomach content analyses?

- *We mention that we used the Bray Curtis Index followed by clustering by UPGMA. However, after the first observation of the cluster dendrogram we notice certain species affect the outcome of the average silhouette width. For this reason, we ran the analysis again excluding these species for which a 100% of diet component is present (Supplementary figure 4). Overall, the defined groups were consistent in the second dendrogram, however, other groups like the Generalists and Crustacean consumers required further assessment. We decided to keep the Generalist group since it shows a wider prey spectrum compared to the Crustacean consumers as shown in the heatmap which now replaces the Supplementary figure 3 and 4. This information is now included in the supplementary methods.*
-

Figures:

-Figure 3A: Perhaps a log scale would facilitate the interpretation. Fig c-f is disproportionately large when considering its relevance. I suggest moving it to supplement and improve a-b as they are hard to read.

- *We changed the scale with a Y axis break to increase the size of the boxplots. Additionally, we did as suggested regarding the Figures 3c-f.*

-Supplementary figures are poorly labeled and have poor quality: Fig S8: hard to see what the colours denote.

- *We improved the quality of the figures.*

-Figs S8-S9: Please explain abbreviations.

- *Done.*

-I think fig. S9 should be in the main text

- *Now it is, Figure 4a.*

Reviewer #3 (Remarks to the Author):

López-Romero et al. assess the evolution of shark jaw morphology in the context of the 'onshore-offshore' diversification gradient, thus shedding light on the environmental controls of species diversity. The authors find support for the hypothesis that sharks constitute a striking exception to the general pattern, demonstrating that high rates of morphological evolution occur not only in reef but also in deep-water habitats.

I congratulate the authors for this interesting piece of work, and especially for tackling this broad-interest question with such a detailed methodological procedure. I think the methodology is robust (but see some comments below) and the conclusions are sound and well supported. Therefore, I strongly support the eventual publication of this article in Communications Biology. I have some comments that the authors might want to consider in reviewing their manuscript. I feel they could have run some extra 'sensitivity' analyses considering uncertainty associated to node calibrations, lifestyle categorizations, and some morphological outliers.

Lines 124-128. Isotopic approaches constitute good alternatives to stomach content analysis. Among many other papers see:

Speed, C. W., et al. "Trophic ecology of reef sharks determined using stable isotopes and telemetry." *Coral reefs* 31.2 (2012): 357-367.

Hussey, Nigel E., et al. "Stable isotope profiles of large marine predators: viable indicators of trophic position, diet, and movement in sharks?." *Canadian Journal of Fisheries and Aquatic Sciences* 68.12 (2011): 2029-2045.

MacNeil, M. Aaron, Gregory B. Skomal, and Aaron T. Fisk. "Stable isotopes from multiple tissues reveal diet switching in sharks." *Marine Ecology Progress Series* 302 (2005): 199-206.

Estrada, James A., et al. "Predicting trophic position in sharks of the north-west Atlantic Ocean using stable isotope analysis." *Journal of the Marine Biological Association of the United Kingdom* 83.6 (2003): 1347-1350.

Speed, C. W., et al. "Trophic ecology of reef sharks determined using stable isotopes and telemetry." *Coral reefs* 31.2 (2012): 357-367.

McCormack, Jeremy, et al. "Trophic position of *Otodus megalodon* and great white sharks through time revealed by zinc isotopes." *Nature Communications* 13.1 (2022): 1-10.

Kast, Emma R., et al. "Cenozoic megatooth sharks occupied extremely high trophic positions." *Science Advances* 8.25 (2022): eabl6529.

- *We do agree with this observation regarding the estimation of trophic level. On the other hand, we were also interested on the prey category for the analysis, that is why we used the percentages and dissimilarity index for these categories. We are not sure if the isotope values can extract a signal associated with a specific type of prey. A paper describing the methods to reveal the prey source states: "... differences in $\delta^{13}\text{C}$ values can distinguish between a consumers reliance on C3 and C4 plants (Peterson & Fry, 1987) or between benthic and pelagic resources (France, 1995). Prey items that are enriched or depleted in ^{15}N , allow separation of for example, C3, C4 plants and phytoplankton (Hayden et al., 2015). Such analyses work well when putative dietary resources can be well established a priori, the isotope ratios of different primary producers are clearly differentiated and the TDF (Trophic Discrimination Factor) is well resolved. Mixing models provide good quantitative estimates of resource flow, however their diet resolution is normally limited to 3–5 diet items depending on the number of isotope tracers, and these models are rarely suitable for depicting individual diet items..."*

Nielsen, J. M., Clare, E. L., Hayden, B., Brett, M. T., & Kratina, P. (2018). Diet tracing in ecology: Method comparison and selection. *Methods in Ecology and Evolution*, 9(2), 278-291.

- *We have included some of these references to indicate other possibilities for the analysis. Thank you for the comment.*

Lines 157-160. Most squaliforms also cluster in a comparatively narrow area of the morphospace (PC1-2, and PC1-3), and only a few taxa (I think Somniosidae and Oxynotidae) occupy most extreme positive values in PC1. Could the results be driven by those "outliers"? Did the authors consider rerunning (at least some) analyses excluding this taxa?

- *We have conducted some analyses excluding both taxa mentioned (the other one is *Isistius brasiliensis*). The exclusion of these did not change drastically the observations on the disparity. In fact, after considering two metrics for disparity (sum of variances and averaged squared distances) the results remain consistent as shown here:*

Lines 351-365. The discussion on the whether or not diversification rate and rates of morphological evolution is confusing and I think it could be better expressed. The authors did not perform any analyses of (taxonomic) diversification rates, how then they tested this? From data in Sorenson et al. (2014)? If so, a figure showing taxonomic diversification rates and morphological change rates would be very welcome (even in the supplementary material).

In addition, the fact that Carchariniiformes in ‘reefs’ display low disparity and morphological evolutionary rates does not entail that diversity and disparity are decoupled in the different environments. Actually, reefs and deep-waters show high speciation rates (see Sorenson et al. 2014) and high morphological disparity. The difference is that in the former only orectolobiformes contribute to this pattern (but it is still true) while, in the second case, apparently all the clades do.

- *We support this observation also with other references, specifically for Squaliformes and Orectolobiformes (Straube et al, 2015; Claes et al, 2015; Corrigan & Behegaray, 2009; Boyd & Seitz, 2021). In this regard, it is out of*

the scope to estimate the taxonomic diversification limited to the number of species for which we have morphological data. One of the limitations is a precise calibration for the tree. A better outcome of such analysis would only be possible with an accurate assignment of some fossils for specific groups. Nevertheless, the random subsample of trees that we used can account for that phylogenetic uncertainty.

- The second point, we tried other possible arrangements of the groups as suggested in the comment Lines 504-507. One of the classifications was the one used in the paper of Martinez et al, (2021), this one show that shallow water species have the lowest disparity (see table below). However, we do disagree with this classification, since the presented one in Martinez et al. considers the maximum depth to classify the groups as “deep”. Alternatively, a classification which considers the medium range depth distribution would make more sense (Andrzejaczek et al, 2022).

	subsets	n	obs	bs.median
Fishbase	Bathy Demersal	15	0.012	0.011
	Bathy Pelagic	2	0	0
	Bentho Pelagic	5	0.003	0.002
	Demersal	28	0.009	0.008
	Pelagic	13	0.006	0.005
	Reef	26	0.01	0.01
Fishbase no low n	Bathy Demersal	15	0.012	0.011
	Demersal	28	0.009	0.008
	Pelagic	13	0.006	0.005
	Reef	26	0.01	0.01
Martinez et al, 2021	Shallow	55	0.024	0.024
	Twilight	20	0.032	0.029
	Deep	13	0.039	0.035

The observed disparity and bootstrapped disparity are shown in the table, as well as the individuals in each group.

Straube, N., Li, C., Claes, J. M., Corrigan, S., & Naylor, G. J. (2015).

Molecular phylogeny of Squaliformes and first occurrence of bioluminescence in sharks. BMC evolutionary biology, 15(1), 1-10.

Claes, J. M., Nilsson, D. E., Mallefet, J., & Straube, N. (2015). *The presence of lateral photophores correlates with increased speciation in deep-sea bioluminescent sharks. Royal Society open science, 2(7), 150219.*

Boyd, B. M., & Seitz, J. C. (2021). *Global shifts in species richness have shaped carpet shark evolution. BMC Ecology and Evolution, 21(1), 1-10.*

Corrigan, S., & Beheregaray, L. B. (2009). *A recent shark radiation: molecular phylogeny, biogeography and speciation of wobbegong sharks (family: Orectolobidae). Molecular Phylogenetics and Evolution, 52(1), 205-216.*

Andrzejaczek, S., Lucas, T. C., Goodman, M. C., Hussey, N. E., Armstrong, A. J., Carlisle, A., ... & Sulikowski, J. A. (2022). Diving into the vertical dimension of elasmobranch movement ecology. *Science advances*, 8(33), eabo1754.

Lines 387-417. As it is, it seems these analyses/results are not contextualized within the general rationale of the paper. I wonder whether the authors could better explain why providing a reconstruction of the ancestral jaw morphology and diets is important for the main goal of this article (i.e., testing hypotheses of diversification gradients and their linking to morphological specialization). In this sense, the authors recognise that the inclusion of *Callorhynchus milli* could have biased their ACSR analyses. I wonder whether they could expand on why the inclusion of an outgroup is important here and why not to include other potentially more suitable (extinct) taxa (closest to the ancestral node of elasmobranch).

*Indeed, extinct chondrichthyans like hybodontiforms and ctenacanthiforms would be certainly be good candidates for further investigation. Unfortunately, complete and undeformed 3D fossil material is very rare and restricted to a few species only. For instance, Lane & Maisey (2012) described 3D preserved jaws of the hybodontiform *Tribodus* from the Lower Cretaceous of Brazil. This taxon, however, is problematic, because its jaw morphology seems to be very derived and rather more similar to that of batoids.*

A major problem currently is that the composition of stem selachians still is controversially discussed. The position of those taxa that are considered to be stem selachians on the stem also is uncertain. This mainly is related to the nature of preservation of fossils and the lack of unambiguous morphological characters identifying these taxa reliably. The motivation for reconstructing the hypothetical ancestral shape of the lower jaw therefore is that it might open new perspectives for identifying basal stem selachians in the future.

*Lane, J. A., & Maisey, J. G. (2012). The visceral skeleton and jaw suspension in the durophagous hybodontid shark *Tribodus limae* from the Lower Cretaceous of Brazil. *Journal of Paleontology*, 86(5), 886-905.*

Line 477. Do the set of 100 trees consider node-calibration uncertainty? This constitute another potential source of uncertainty that should be accounted for.

- *The subsampled trees consider node calibration uncertainty. In the updated figure 4 (now figure 5) the disparity through time shows the position of each node at each time for the 100 trees selected.*

Lines 485 and 499. Tables S1 and S2 do not contain this information. Do the authors mean Data S1 and S2?

- *Yes, thanks for the observation, correction done.*

Lines 487-488. How did the authors determine the number of clusters for establishing dietary groups? The authors might better specify which are the criteria they followed for this.

- *We address this point (see Reviewer's 2 comment).*
- *We mention that we used the Bray Curtis Index followed by clustering by UPGMA. However, after the first observation of the cluster dendrogram we notice certain species affect the outcome of the average silhouette width. For this reason, we ran the analysis again excluding these species for which a 100% of diet component is present (Supplementary figure 4). Overall, the defined groups were consistent in the second dendrogram, however, other groups like the Generalists and Crustacean consumers required further assessment. We decided to keep the Generalist group since it shows a wider prey spectrum compared to the Crustacean consumers as shown in the heatmap which now replaces the Supplementary figure 3 and 4. This information is now included in the supplementary methods.*

Lines 504-507. I found the terminology used for categorizing lifestyles/habitats confusing. Actually, several of the species categorized as pelagic inhabit mostly the continental shelf. I would say the "shelf" species here constitute actually demersal (or benthopelagic) sharks. A neritic species ('shelf' species) could be either demersal or pelagic; and a pelagic species could be either neritic or oceanic.

- *We address this issue in the Lines 351-365 comment.*

Lines 504-507. Did the authors consider alternative categorization of the species in order to account for potential uncertainty in their classification in lifestyles? This would be advisable as the classification of some species is certainly non trivial.

- *The possible classifications were not considered at first. However, we ran the same analyses with alternative ones for lifestyles in Lines 351-365 comment. The outcome of these analyses does not change the main results regarding disparity. In the case of alternative classification used by Martinez et al (2021), the ranges of depth for sharks are too variable. The study of Martinez et al uses maximum depth data, which can be misleading when assigning a classification. In this case we preferred to use the data on median range distribution. Under this classification we obtained a similar result as the one from Martinez et al.*

Lines 510-512. The employment of minimum bending energy to slide semilandmarks is not arbitrary and should be justified in the main text/supplementary methods.

- *Reaction changed: “surface semilandmarks are allowed to slide in order to minimize the bending energy and to avoid semilandmarks passing across an anatomical landmark (Gunz & Mitteroecker, 2013).”*

Please revise the citations to supplementary files in the Supplementary Information Methods. I found several inconsistencies.

We revised the inconsistencies. All citations should be in order now.

REVIEWERS' COMMENTS:

Reviewer #1 (Remarks to the Author):

This is a very strong revision of a manuscript that I reviewed previously. It is a super-interesting study of the diversification of shark jaws, using 3D landmark morphometric data from CT scans. A very extensive sequence of comparative phylogenetic methods are employed in the study. This makes for a complicated set of results – showing, clade-specific, diet-specific, and habitat-specific disparity and rates of jaw evolution. The study shows that rates of jaw shape evolution are highest in the deep sea – an interesting result that parallels recent findings regarding body shape in marine teleost fishes. The authors have done a very good job with this revision of clarifying many things I found confusing the first time around and I find the manuscript more straight-forward and easier to comprehend. I have some relatively minor comments that I strongly recommend the authors address, but my feeling is that this manuscript is in good shape now, and relatively little additional revision is needed.

L37. I guess I do not understand this statement that the rates of jaw evolution corroborates a pattern of even diversification across habitats? Figures 3a and 3b seem to show strong support that disparity and rate of jaw evolution are not the same in different habitats. I feel like I must be misunderstanding the statement in the abstract, since this is major result in the paper.

L46. Given the ambiguity in the results of the Sorenson et al 2014 paper on speciation rate, would it be better to say here: ‘...evolutionary rates of jaw evolution are associated with a tendency toward higher rates of speciation in deep water, but not on reefs.’?

L130 Delete the first instance of ‘roughly’ in this line.

L139. Use ‘related to’ instead of ‘relate to’.

L140. Use ‘tapers’ instead of ‘taper’.

L224. It might be helpful to indicate the rates for the slower habitats so the reader can understand how big the difference is. This could be done by reporting those average values of the rate parameter or by describing the rate of the each higher rate as a ratio over the lower habitat rate. In general, would it be helpful to express the rate estimates as a ratio with the category with the lowest rate? This would make the values multiples of the slowest category and facilitates comparison (deep sea is X.X times faster than open ocean)?

L307-308. Use ‘in-lever’ instead of ‘in-level’, and ‘out-lever’ instead of ‘out-level’.

L365. Use ‘supported’ instead of ‘support’.

L499. The study includes landmark data from 145 specimens from 90 species. How were these 145 specimens mapped to the phylogeny of 90 species? Were average shapes calculated per species? Explain how this was done.

L535. Use ‘phylogenetically aligned PCA’ instead of ‘phylogenetic aligned components analysis’.

L568, L276 & Figure 5C. As I mentioned in my previous review, I believe the disparity through time plot is possibly being misinterpreted. L568 states that the `dispRity` package of Guillaume, 2018 was used for this. However, `dttdispRity` is a wrapper for the `Geiger::dttdispRity` function, which is what appears to have been used to produce the plots in Fig 5C and in the author’s response document. These plots actually show AVERAGE SUBCLADE DISPARITY EXPRESSED AS A FRACTION OF TOTAL CLADE DISPARITY, plotted through time. Thus, the line slopes downward through time since younger and younger clades have less average disparity. The main way these are interpreted is when the empirical line moves outside of the 95% band of Brownian simulations. Thus, the plot in Figure 5 is consistent with Brownian motion from the base of the tree until about 120 mya when the line goes above the Brownian band. This indicates a period starting about 120 mya during which clades established in this period achieve greater disparity than expected under constant rate

Brownian motion. This could be evidence of a period of increased rate but there are other possible explanations.

L577. Use 'phylogenetically aligned PCA' instead of 'phylogenetic-aligned PCA'.

L577. Please report the number of PC axes retained using this process.

L580. Use 'trait' instead of 'traits'.

L583 & L584. Use 'chains' instead of 'chains'.

L593. I can see how using 100 simmaps would account for uncertainty in the evolutionary history of the discrete trait, but I do not think it allows you to account for phylogenetic uncertainty, as stated here. For that, you would need to repeat analyses on a sample of trees from the posterior distribution. Or am I misunderstanding?

L948. Use 'displayed by coloured points' instead of 'displayed by colour points'.

Figure 1. The figure legend indicates there are 19 squaliform specimens but I only see 18 points in the plot? Perhaps one point is obscured by others?

Reviewer #2 (Remarks to the Author):

The authors have provided a revised version of their manuscript, which addresses some of my concerns. However, it seems to me that some of my comments were not carefully considered.

For example, I mentioned that when describing their results on mean evolutionary rates, they say that Orectolobiformes has the highest. This is not the case. Indeed, this order has an outlier value that is much higher than other orders, but it clearly does not have the highest *MEAN* value. In their response, they say that what makes the difference between orders is the outliers, but there is not an analytical/statistical reason for this interpretation, in my opinion. If the authors are interested in outliers and maximum values then they need to explain why and be clear about it. Perhaps that is even a totally different thing.

Similarly, my comment about how body size was factored in phylogenetic patterns results is somewhat addressed in their rebuttal letter, but not really in the manuscript.

Finally, my comment on sampling differences was also not fully addressed. The authors say that they chose orders with more than 3 species, but that only tells me that there is a minimum of species used, and not how *sampling differences* are accounted for, or whether it insert a bias or not.

Reviewer #3 (Remarks to the Author):

I have gone through the rebuttal and the revised manuscript. The authors have addressed all my points and I have no more major concerns to be mentioned.

REVIEWERS' COMMENTS:

Reviewer #1 (Remarks to the Author):

This is a very strong revision of a manuscript that I reviewed previously. It is a super-interesting study of the diversification of shark jaws, using 3D landmark morphometric data from CT scans. A very extensive sequence of comparative phylogenetic methods are employed in the study. This makes for a complicated set of results – showing, clade-specific, diet-specific, and habitat-specific disparity and rates of jaw evolution. The study shows that rates of jaw shape evolution are highest in the deep sea – an interesting result that parallels recent findings regarding body shape in marine teleost fishes. The authors have done a very good job with this revision of clarifying many things I found confusing the first time around and I find the manuscript more straight-forward and easier to comprehend. I have some relatively minor comments that I strongly recommend the authors address, but my feeling is that this manuscript is in good shape now, and relatively little additional revision is needed.

L37. I guess I do not understand this statement that the rates of jaw evolution corroborates a pattern of even diversification across habitats? Figures 3a and 3b seem to show strong support that disparity and rate of jaw evolution are not the same in different habitats. I feel like I must be misunderstanding the statement in the abstract, since this is major result in the paper.

Correction here, it should be “uneven diversification across habitats”. We changed the redaction.

L46. Given the ambiguity in the results of the Sorenson et al 2014 paper on speciation rate, would it be better to say here: ‘...evolutionary rates of jaw evolution are associated with a tendency toward higher rates of speciation in deep water, but not on reefs.’?

Done as suggested.

L130 Delete the first instance of ‘roughly’ in this line.

Done.

L139. Use ‘related to’ instead of ‘relate to’.

Done.

L140. Use 'tapers' instead of 'taper'.

Done.

L224. It might be helpful to indicate the rates for the slower habitats so the reader can understand how big the difference is. This could be done by reporting those average values of the rate parameter or by describing the rate of the each higher rate as a ratio over the lower habitat rate. In general, would it be helpful to express the rate estimates as a ratio with the category with the lowest rate? This would make the values multiples of the slowest category and facilitates comparison (deep sea is X.X times faster than open ocean)?

Now the values are reported as fold change, and we also include a supplementary table (S5) to display the mean, median, sd, and se for each category.

L307-308. Use 'in-lever' instead of 'in-level', and 'out-lever' instead of out-level'.

Done

L365. Use 'supported' instead of 'support'.

Done

L499. The study includes landmark data from 145 specimens from 90 species. How were these 145 specimens mapped to the phylogeny of 90 species? Were average shapes calculated per species? Explain how this was done.

Indeed, this was done, we indicate this in the Methods, Lines 517-518:

"using the averaged by species Procrustes aligned landmark coordinates"

We just take the coordinates in a two dimensional array and apply the mean function base in R.

L535. Use 'phylogenetically aligned PCA' instead of phylogenetic aligned components analysis'.

Done

L568, L276 & Figure 5C. As I mentioned in my previous review, I believe the disparity through time plot is possibly being misinterpreted. L568 states that the dispRity package of Guillerme, 2018 was used for this. However, dtt.dispRity is a wrapper for the Geiger::dtt function, which is what appears to have been used to produce the plots in Fig 5C and in the author's response document. These plots actually show AVERAGE SUBCLADE DISPARITY EXPRESSED AS A FRACTION OF TOTAL CLADE DISPARITY, plotted through

time. Thus, the line slopes downward through time since younger and younger clades have less average disparity. The main way these are interpreted is when the empirical line moves outside of the 95% band of Brownian simulations. Thus, the plot in Figure 5 is consistent with Brownian motion from the base of the tree until about 120 mya when the line goes above the Brownian band. This indicates a period starting about 120 mya during which clades established in this period achieve greater disparity than expected under constant rate Brownian motion. This could be evidence of a period of increased rate but there are other possible explanations.

Indeed it is a wrapper of Geiger::dtc, with the difference that in dtc.dispRity it is possible to change the disparity metric, to be consistent with the metric we used for the other analyses (sum of variances with all the landmark data). Interestingly, as the reviewer mentions, this period is consistent with the highest diversity found in palaeontological studies (Guinot et al, 2012; Guinot & Cavin, 2016; Guinot & Cavin, 2020). Particularly, Guinot and Cavin (2020) argue that elasmobranch diversification in the past is correlated with continental fragmentation. We include briefly in lines 379, 380 the relationship of this disparity peak also observed in palaeodiversity studies.

“This can be detected in the disparity through time analysis, which is consistent with the increase in diversity through the fossil record (from the late Jurassic to late Cretaceous) of sharks observed (Guinot et al, 2012; Guinot and Cavin, 2020).”

Guinot, G., Adnet, S., & Cappetta, H. (2012). An analytical approach for estimating fossil record and diversification events in sharks, skates and rays. *Plos one*, 7(9), e44632-e44632.

Guinot, G., & Cavin, L. (2016). ‘Fish’ (Actinopterygii and Elasmobranchii) diversification patterns through deep time. *Biological Reviews*, 91(4), 950-981.

Guinot, G., & Cavin, L. (2020). Distinct responses of elasmobranchs and ray-finned fishes to long-term global change. *Frontiers in Ecology and Evolution*, 7, 513.

L577. Use ‘phylogenetically aligned PCA’ instead of ‘phylogenetic-aligned PCA.

Done.

L577. Please report the number of PC axes retained using this process.

Done.

L580. Use ‘trait’ instead of ‘traits’.

Done.

L583 & L584. Use chains' instead of chains.

Done.

L593. I can see how using 100 simmaps would account for uncertainty in the evolutionary history of the discrete trait, but I do not think it allows you to account for phylogenetic uncertainty, as stated here. For that, you would need to repeat analyses on a sample of trees from the posterior distribution. Or am I misunderstanding?

That is exactly what we did. We used the 100 trees (with different topologies) from the posterior distribution. We explain this part now in the Methods section:

“With the selected model for each category, we made a stochastic character mapping on the 100 trees subsample using the `make.simmap` function in the `phytools` package (Revell, 2012). Afterwards, we used the character mapped trees and the aligned landmark coordinates averaged by species with the function `mvgl` in the package `mvMorph` under a Brownian Motion model (Clavel et al, 2015). We extracted the model parameter estimates to obtain the evolutionary rates as described in Fabre et al. (2021).”

L948. Use ‘displayed by coloured points’ instead of ‘displayed by colour points’.

Done.

Figure 1. The figure legend indicates there are 19 squaliform specimens but I only see 18 points in the plot? Perhaps one point is obscured by others?

*The other one in the extreme positive PC1 is overlapped by another specimen, this corresponds to *Oxynotus centrina*. For this, we included the name labels in the supplementary figure S6.*

Reviewer #2 (Remarks to the Author):

The authors have provided a revised version of their manuscript, which addresses some of my concerns. However, it seems to me that some of my comments were not carefully considered.

For example, I mentioned that when describing their results on mean evolutionary rates, they say that Orectolobiformes has the highest. This is not the case. Indeed, this order has an outlier value that is much higher than other orders, but it clearly does not have the highest *MEAN* value. In their response, they say that what makes the difference between orders is the outliers, but there is not an analytical/statistical reason for this interpretation,

in my opinion. If the authors are interested in outliers and maximum values then they need to explain why and be clear about it. Perhaps that is even a totally different thing.

It does have the highest mean value. In the previous rebuttal letter, we indicated that to observe more clearly this, we made a table with the mean, median, sd and se of the results of the 100 random trees (supplementary table S5). In the case of the figure, it is usual to observe in the boxplots the median value. Because of this, now we include a mean value in the boxplots to highlight these differences. Also, following the recommendation of Reviewer 1, we report some of the values in the text as fold-change. And we rephrase the previous reply, since from the different trees used to obtain these results, some of the observed topologies there are some observations that some of the families within Orectolobiformes are the result of a recent radiation. We mention this in lines 346-347:

“Additionally, the diversification pattern for galeomorphs in reefs is supported by a rather recent diversification event in orectolobiforms (Corrigan, & Beheregaray, 2009), which also is support by the fossil record of orectolobiforms (Boyd, & Seitz, 2021).”

Similarly, my comment about how body size was factored in phylogenetic patterns results is somewhat addressed in their rebuttal letter, but not really in the manuscript.

We mentioned in the previous reply, and it was included in the supplementary information:

“A phylogenetic MACOVA of the shape and centroid size interaction suggest that size is significant but contributes little to the shape variation ($r^2 = 0.04085$; $F = 3.705$; $Z = 2.3226$; $p = 0.027$). Therefore, we did not correct for size using the residuals of the shape regressed on the centroid size. We have included this information in the supplementary information regarding the methods.”

But for clarity we include this information in the methods instead of the supplementary material. Now this information is in lines 518-520

Finally, my comment on sampling differences was also not fully addressed. The authors say that they chose orders with more than 3 species, but that only tells me that there is a minimum of species used, and not how *sampling differences* are accounted for, or whether it insert a bias or not.

Due to the availability of some specimens in the collections there is this limitation, we acknowledge that fact. One common procedure to overcome this issue is through bootstrapping resample (which was done for the disparity analysis) mentioned in the corresponding methods section (line 528 of the manuscript) and also through permutations (implemented in the manova.gls function), in the same way mentioned

in its section (line 544). Perhaps the statement got lost within the other Reviewer's (#1) comment, who raised the same issue. In that instance we replied:

Regarding the sample size, that is why the presented results are bootstrapped, which is a common procedure for such instances. Of course, a comparison of samples with only one individual would be meaningless for the purpose of variance, like in the case of Echinorhiniformes and Pristiphoriformes, for which we only had one species available.

We now include a short sentence in the corresponding lines of the methods.

Reviewer #3 (Remarks to the Author):

I have gone through the rebuttal and the revised manuscript. The authors have addressed all my points and I have no more major concerns to be mentioned.

Thank you so much!